# Calcium influx through CRAC channels controls actin organization and dynamics at the immune synapse

**Catherine A Hartzell[1,2]\*, Katarzyna I Jankowska[3,4], Janis K Burkhardt[3,4], Richard S Lewis[1,2]\***

[1]Immunology Program, Stanford University, Stanford, United States; [2]Department of Molecular and Cellular Physiology, Stanford University, Stanford, United States; [3]Department of Pathology and Laboratory Medicine, Children's Hospital of Philadelphia Research Institute, Philadelphia, United States; [4]Department of Pathology and Laboratory Medicine, Perelman School of Medicine, University of Pennsylvania, Philadelphia, United States

**\*For correspondence:** chartzel@ alumni.stanford.edu (CAH); rslewis@stanford.edu (RSL)

**Competing interests:** The authors declare that no competing interests exist.

**Abstract** T cell receptor (TCR) engagement opens $Ca^{2+}$ release-activated $Ca^{2+}$ (CRAC) channels and triggers formation of an immune synapse between T cells and antigen-presenting cells. At the synapse, actin reorganizes into a concentric lamellipod and lamella with retrograde actin flow that helps regulate the intensity and duration of TCR signaling. We find that $Ca^{2+}$ influx is required to drive actin organization and dynamics at the synapse. Calcium acts by promoting actin depolymerization and localizing actin polymerization and the actin nucleation promotion factor WAVE2 to the periphery of the lamellipod while suppressing polymerization elsewhere. $Ca^{2+}$-dependent retrograde actin flow corrals ER tubule extensions and STIM1/Orai1 complexes to the synapse center, creating a self-organizing process for CRAC channel localization. Our results demonstrate a new role for $Ca^{2+}$ as a critical regulator of actin organization and dynamics at the synapse, and reveal potential feedback loops through which $Ca^{2+}$ influx may modulate TCR signaling.

## Introduction

Soon after a T cell encounters cognate antigen on an antigen-presenting cell (APC), it spreads out over the cell's surface, forming a tightly apposed structure known as the immune synapse (*Bromley et al., 2001*; *Yokosuka and Saito, 2010*; *Dustin, 2008*). The synapse regulates T cell activation by maximizing the contact area and organizing the T cell receptors (TCR) and associated signaling proteins into zones. Strong antigenic stimuli create three concentric regions (*Monks et al., 1998*; *Grakoui et al., 1999*): a central supramolecular activation cluster (cSMAC), an intermediate zone (the peripheral SMAC, or pSMAC), and a zone at the synapse edge (the distal SMAC, or dSMAC) (*Freiberg et al., 2002*). TCRs assemble with scaffolding and signaling proteins to form microclusters in the dSMAC which migrate centripetally towards the cSMAC (*Grakoui et al., 1999*; *Krummel et al., 2000*; *Campi et al., 2005*; *Varma et al., 2006*; *Yokosuka et al., 2005*). As they move, TCR microclusters activate a MAP kinase cascade and $Ca^{2+}$ influx through $Ca^{2+}$ release-activated $Ca^{2+}$ (CRAC) channels, both of which are essential to initiate gene expression programs that drive T cell proliferation and differentiation (*Feske et al., 2001*). Signaling by TCR microclusters is terminated as they enter the cSMAC by the dissociation of signaling proteins (*Yokosuka et al., 2005*; *Campi et al., 2005*; *Varma et al., 2006*) and endocytosis of TCRs (*Lee et al., 2003*; *Liu et al., 2000*; *Das et al., 2004*). Thus, the strength of signaling at the synapse is thought to reflect a

**eLife digest** An effective immune response requires the immune system to rapidly recognize and respond to foreign invaders. Immune cells known as T cells recognize infection through a protein on their surface known as the T cell receptor. The T cell receptor binds to foreign proteins displayed on the surface of other cells. This interaction initiates a chain of events, including the opening of calcium channels embedded in the T cell membrane known as CRAC channels, which allows calcium ions to flow into the cell. These events ultimately lead to the activation of the T cell, enabling it to mount an immune response against the foreign invader.

As part of the activation process, the T cell spreads over the surface of the cell that is displaying foreign proteins to form an extensive interface known as an immune synapse. The movement of the T cell's internal skeleton (the cytoskeleton) is crucial for the formation and function of the synapse. Actin filaments, a key component of the cytoskeleton, flow from the edge of the synapse toward the center; these rearrangements of the actin cytoskeleton help to transport clusters of T cell receptors to the center of the synapse and enable the T cell receptors to transmit signals that lead to the T cell being activated. It is not entirely clear how the binding of T cell receptors to foreign proteins drives the actin rearrangements, but there is indirect evidence suggesting that calcium ions may be involved.

Hartzell et al. have now investigated the interactions between calcium and the actin cytoskeleton at the immune synapse in human T cells. T cells were placed on glass so that they formed immune synapse-like connections with the surface, and actin movements at the synapse were visualized using a specialized type of fluorescence microscopy. When calcium ions were prevented from entering the T cell, the movement of actin stopped almost entirely. Thus, the flow of calcium ions into the T cell through CRAC channels is essential for driving the actin movements that underlie immune synapse development and T cell activation.

In further experiments, Hartzell et al. tracked the movements of CRAC channels and actin at the synapse and found that actin filaments create a constricting "corral" that concentrates CRAC channels in the center of the synapse. Thus, by driving cytoskeleton movement, calcium ions also help to organize calcium channels at the immune synapse. Future work will focus on identifying the actin remodeling proteins that enable calcium ions to control this process.

dynamic balance between formation of new microclusters in the dSMAC/pSMAC and their disassembly and internalization in the cSMAC.

Actin reorganization at the synapse is crucial for TCR microcluster assembly, movement and signaling (*Babich et al., 2012*; *Campi et al., 2005*; *Delon et al., 1998*; *Kaizuka et al., 2007*; *Liu et al., 1995*; *Valitutti et al., 1995*; *Varma et al., 2006*; *Yi et al., 2012*; *Kumari et al., 2015*). In the dSMAC, actin is dense and highly branched (*Parsey and Lewis, 1993*; *Bunnell et al., 2001*) and exhibits rapid retrograde flow similar to actin in the lamellipod of migrating cells. In the neighboring pSMAC region, actin is less dense and resembles a lamella with actin organized into concentric arcs by myosin IIA (*Babich et al., 2012*; *Yi et al., 2012*; *Yu et al., 2012*). Actin is sparse in the actin-depleted zone (ADZ) corresponding to the cSMAC. Centripetal actin flow regulates TCR function in at least two ways. First, it transports TCR microclusters towards the cSMAC where they are disassembled, limiting the signaling lifetime of each microcluster to a few minutes (*Yu et al., 2010*; *Varma et al., 2006*; *Yokosuka et al., 2005*). Second, actin polymerization and depolymerization are critical for microcluster formation and function, based on the ability of cytochalasin D (an actin polymerization inhibitor) and jasplakinolide (an actin depolymerization inhibitor) to rapidly quell microcluster formation, MAP kinase signaling, and $Ca^{2+}$ influx at the synapse (*Valitutti et al., 1995*; *Varma et al., 2006*; *Rivas et al., 2004*; *Babich et al., 2012*; *Yi et al., 2012*). Thus, the mechanisms that control actin organization and flow at the synapse are key to understanding synapse formation as well as T-cell signaling.

TCR stimulation is known to drive actin reorganization by activating the Rho-family GTPases Rac1 and Cdc42, which function via Wiscott-Aldrich syndrome protein (WASp) and WASp-family verprolin homologous protein (WAVE2) to initiate actin nucleation through the Arp2/3 complex

(*Billadeau et al., 2007*). Recent studies have shown that actin polymerization collaborates with myosin IIA contractility to drive retrograde actin flow from the lamellipod to the ADZ, although there is some disagreement as to their relative contributions (*Babich et al., 2012*; *Yi et al., 2012*). The mechanisms that control retrograde flow at the synapse are still not fully understood, and the possibility remains that a master regulator of some kind may act on a global scale to organize this process.

Indirect evidence suggests that intracellular $Ca^{2+}$ may regulate actin organization and dynamics at the synapse. Elevated intracellular $Ca^{2+}$ ([$Ca^{2+}$]$_i$) in T cells has been associated with such cytoskeleton-dependent processes as motility arrest (*Negulescu et al., 1996*; *Bhakta et al., 2005*), cell rounding (*Donnadieu et al., 1994*), cell spreading (*Bunnell et al., 2001*) and synapse stabilization (*Negulescu et al., 1996*; *Krummel et al., 2000*; *Delon et al., 1998*). In addition, T cells express a range of $Ca^{2+}$-sensitive proteins known to regulate actin depolymerization, severing, bundling, and capping (*Babich and Burkhardt, 2013*; *Joseph et al., 2014*; *Janmey, 1994*). TCR engagement is known to elicit $Ca^{2+}$ influx through CRAC channels via a cascade in which PLCγ generates inositol 1,4,5-trisphosphate (IP$_3$), releasing $Ca^{2+}$ from the ER and causing the ER $Ca^{2+}$ sensor STIM1 to redistribute to ER-plasma membrane (PM) junctions (*Wu et al., 2006*; *Luik et al., 2006*) where it traps and activates Orai1, the pore-forming subunit of the CRAC channel (*Luik et al., 2006*; *Wu et al., 2014*). STIM1 and Orai1 colocalize at early times at the immune synapse (*Lioudyno et al., 2008*; *Barr et al., 2008*) and later at the distal pole of the cell (*Barr et al., 2008*), but functional CRAC channel complexes as indicated by $Ca^{2+}$ influx have only been shown at the synaptic contact zone (*Lioudyno et al., 2008*). The precise localization of CRAC channels at the synapse, the mechanisms that control their localization, and their possible effects on actin organization and dynamics are all unknown.

In this study, we applied an in vitro model system to investigate the localization of CRAC channels and the role these channels may play in regulating the actin cytoskeleton at the immune synapse. We found that $Ca^{2+}$ influx through CRAC channels acts at multiple levels to organize actin and promote retrograde flow, which in turn drives ER remodeling and the localization of STIM1 and Orai1 to the center of the synapse. In this way, $Ca^{2+}$ self-organizes CRAC channels at the synapse while creating feedback loops that may help regulate T cell sensitivity to antigen.

## Results

### STIM1 and Orai1 accumulate in the actin-depleted zone of the synapse

To study the location and redistribution of the population of STIM1/Orai1 complexes positioned at the synapse, Jurkat T cells expressing STIM1 labeled with mCherry (mCh-STIM1) and Orai1 labeled with EGFP (Orai1-EGFP) were stimulated on coverslips coated with anti-CD3 mAb (*Bunnell et al., 2001*). Under these conditions, the cells spread over the coverslip to form a structure resembling an immune synapse and time-lapse TIRF microscopy can be used to obtain high resolution 2-dimensional images of the cell region within 200 nm of the coverslip. Previous studies have shown that cells stimulated in this way reorganize their cytoskeleton similarly to T cells forming conjugates with APCs or binding to peptide-MHC complexes in supported planar bilayers (*Parsey and Lewis, 1993*; *Bunnell et al., 2001*; *Yi et al., 2012*).

After settling on stimulatory coverslips, cells spread over several minutes until they reached a constant size and roughly circular shape. Puncta containing STIM1 and Orai1 appeared at the contact zone beginning within seconds of initial contact and continuing through the spreading phase. After cells had spread fully (3–7 min after contact with the coverslip), colocalized puncta of STIM1 and Orai1 continued to increase in number and intensity over the next several minutes and appeared to be confined to the center of the synapse (68 of 82 cells; *Figure 1A* and *Video 1*). While the great majority of cells had centralized puncta, the abundance varied from only 5 to an array too densely packed to accurately count, possibly reflecting cell-to-cell variations in STIM1 and Orai1 expression and the degree of ER [$Ca^{2+}$] depletion. In a minority of cells (27 of 68 cells), puncta containing STIM1 and Orai1 were detected near the periphery and moved toward the center of the synapse (*Figure 1B,C*, *Video 1*) with an average velocity of 47 ± 3 nm/s (n = 24 puncta; mean ± SEM). These motile puncta were more frequently detected in 0.5–0.8 mM extracellular $Ca^{2+}$ ($Ca^{2+}_o$; 48% of 33 cells) than in 2 mM $Ca^{2+}_o$ (31% of 35 cells), probably because a greater degree of ER $Ca^{2+}$ depletion

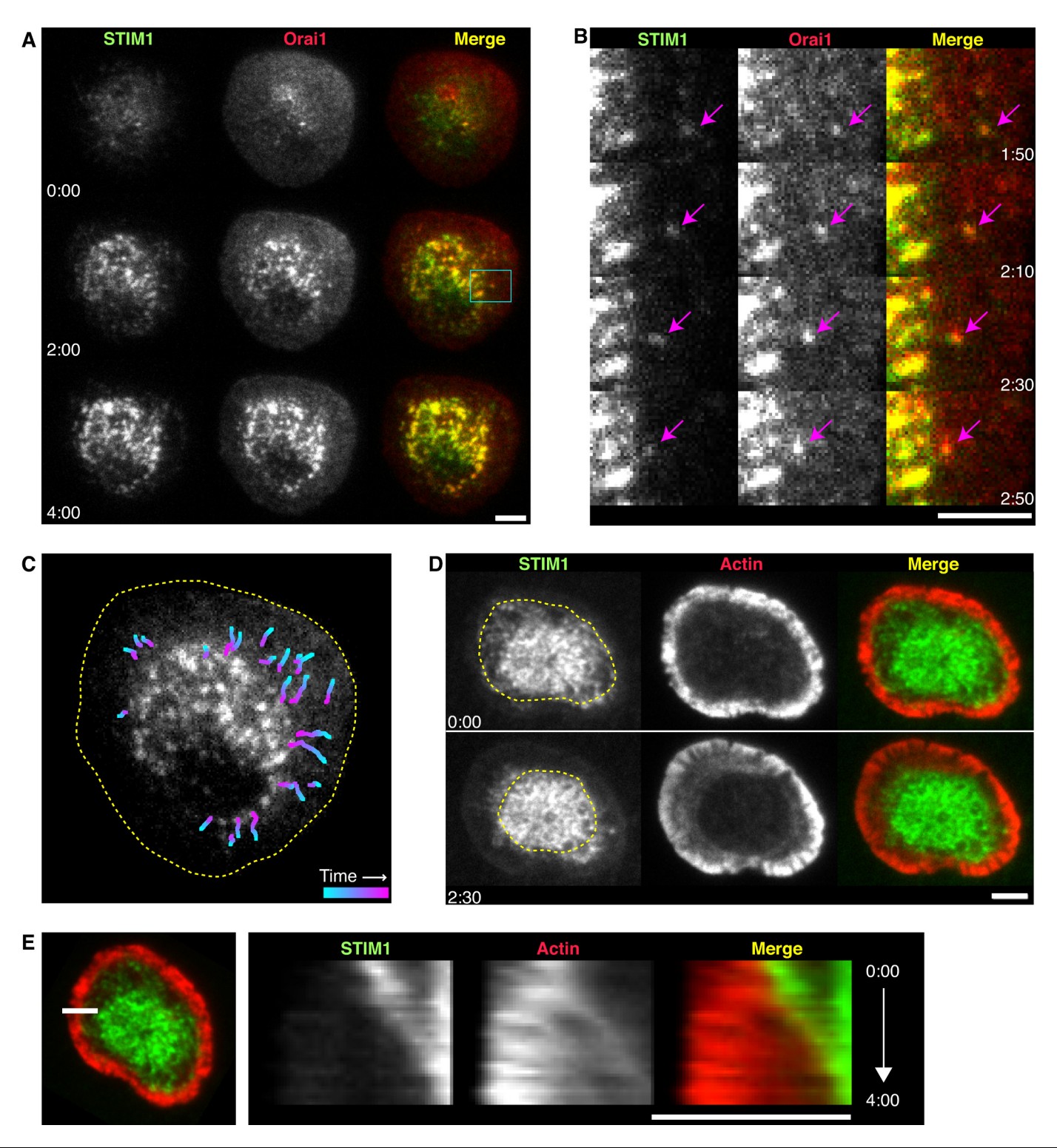

**Figure 1.** STIM1 and Orai1 accumulate in puncta in the actin-depleted zone of the immune synapse. (**A**) TIRF images of Jurkat cells stimulated on anti-CD3 coated coverslips in 0.8 mM $Ca^{2+}_o$. mCh-STIM1 (green) and Orai1-EGFP (red) puncta accumulate in the center of the synapse over time. Images taken from *Video 1*. Scale bar, 5 µm. (**B**) Magnification of the boxed region in **A** shows a STIM1/Orai1 punctum (arrows) moving toward the center of the synapse. Gamma was adjusted to highlight puncta (mCh-STIM1 gamma = 1.3 and Orai1-EGFP gamma = 1.5). (**C**) Centripetal trajectories of STIM1 and Orai1 puncta overlaid on a single image of Orai1-EGFP. The frame-to-frame punctum velocity was 47 ± 3 nm/s (n = 24 particles, mean ± SEM).

*Figure 1 continued on next page*

*Figure 1 continued*

Dashed line indicates the cell edge. (D) ER tubules containing mCh-STIM1 (green) move centripetally with contraction of the EGFP-actin (red) ring. The dashed line indicates the boundary of the ADZ. (E) Kymograph analysis along the indicated line (left) from the cell in D (see *Video 2*). STIM1 moves at the same velocity as the edge of the actin ring. In all panels, time after initial image acquisition is indicated in min:sec; scale bar, 5 μm.

is expected under the reduced $[Ca^{2+}]_o$ conditions. Thus, we suspect that our experiments actually underestimate the number of motile STIM1-Orai complexes at the synapse because they are dim and difficult to detect when ER $[Ca^{2+}]$ is only partially depleted. Puncta of colocalized STIM1 and Orai1 correspond to ER-PM junctions where STIM1-bound Orai1 conducts $Ca^{2+}$ into the cell (*Luik et al., 2006*; *Wu et al., 2006*). Thus, our results suggest that as the synapse matures ER-PM junctions become concentrated in the center of the contact zone, and individual ER-PM junctions loaded with STIM1 and Orai1 translocate from the periphery to further increase the density of $Ca^{2+}$ influx sites in the center.

To address the possible role of actin in the localization of STIM1 and Orai1 puncta, we examined STIM1 and actin dynamics simultaneously in cells expressing mCh-STIM1 and actin labeled with GFP (GFP-actin). In agreement with previous reports (*Bunnell et al., 2001*; *Kaizuka et al., 2007*; *Yu et al., 2010*; *Babich et al., 2012*; *Yi et al., 2012*), cells formed a peripheral lamellipod characterized by a bright band of actin that appeared striated and ruffled in and out of the TIRF plane. At the inner edge of the lamellipod, actin density dropped off sharply, marking the transition into the lamella region where actin formed arc-like structures encircling a central ADZ (*Video 2*). Actin moved continually in a radial retrograde direction at velocities that declined from ~100 nm/s at the cell edge to near 0 nm/s at the border of the ADZ (data not shown). The highest density of STIM1 puncta occurred within the ADZ while dimmer, more dynamic STIM1-containing tubules extended into the lamella (21 of 21 cells; *Figure 1D* and *Video 2*). Kymograph analysis shows that STIM1 puncta in the lamella move centripetally with and at the same velocity as F-actin (*Figure 1E*). These observations suggest that the advancing actin cytoskeletal network moves STIM1/Orai1 puncta and the associated ER-PM junctions towards the ADZ.

## Synaptic ER tubules extend from the ADZ on microtubules but are retrieved by centripetal actin flow

ER organization and behavior at the immune synapse has not been well studied. To better understand the mechanisms underlying CRAC channel positioning we examined ER localization and dynamics and their potential links to the actin cytoskeleton. We labeled actin and the ER membrane by expressing GFP-actin and mCherry tail-anchored to the ER membrane (ER-mCh) (*Bulbarelli et al., 2002*). The ER appeared in the TIRF evanescent field within minutes of cell contact with the stimulatory coverslips and expanded peripherally as cells spread (*Figure 2A* and *Video 3*). The ER near the PM was highly enriched in the ADZ (20 of 20 cells) in both tubular and sheet-like structures that became more centrally concentrated and immobile as the ring of actin surrounding the ADZ contracted. Dynamic ER tubules extended from the ADZ toward the lamellipod (16 of 20 cells;

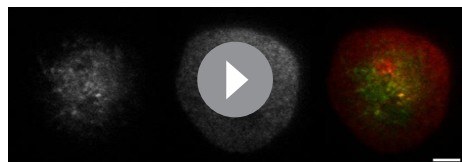

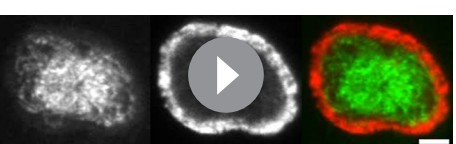

**Video 1.** STIM1 and Orai1 accumulate in puncta in the center of the synapse. Time-lapse TIRF movie of a Jurkat cell expressing mCh-STIM1 (left) and Orai1-EGFP (center) stimulated on an anti-CD3-coated coverslip. A merge of the STIM1 (green) and Orai1 (red) channels is shown at right. Images acquired every 5 s and time compressed 35x. Scale bar, 5 μm. This video supplements *Figure 1A*.

**Video 2.** STIM1 puncta accumulate in the ADZ of the synapse. Time-lapse TIRF movie of a Jurkat cell expressing mCh-STIM1 (left) and GFP-actin (center) stimulated on an anti-CD3-coated coverslip. A merge of the STIM1 (green) and actin (red) channels is shown at right. Images acquired every 5 s and time compressed, 35x. Scale bar, 5 μm. This video supplements *Figure 1D and E*.

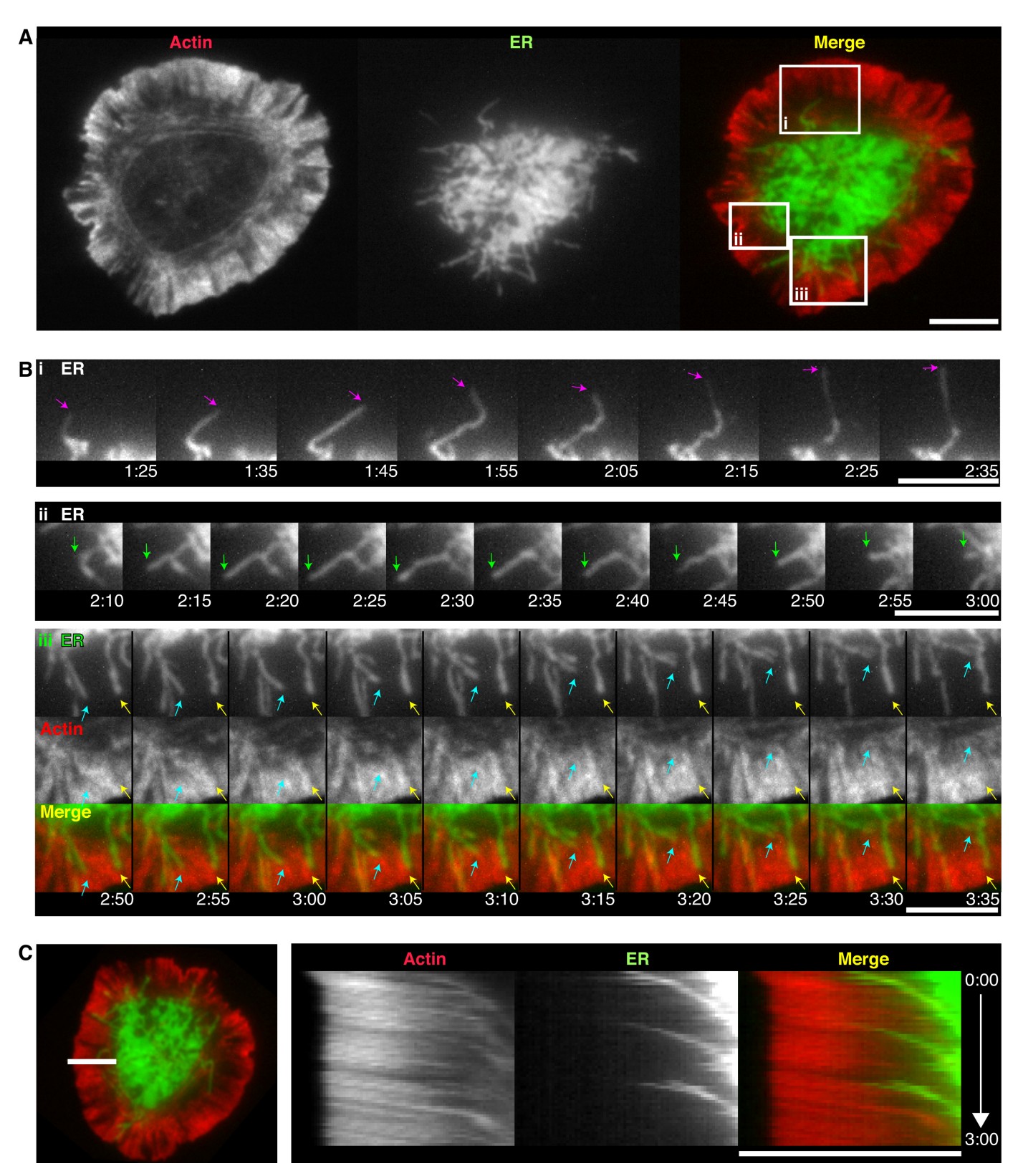

**Figure 2.** Synaptic ER tubules extend from the ADZ and are moved centripetally by actin. (A) TIRF images of a Jurkat cell coexpressing GFP-actin (red) and ER-mCh (green), after spreading on an anti-CD3-coated coverslip. (B) Magnification of the boxed regions in A depicting an extending ER tubule (i,

*Figure 2 continued on next page*

*Figure 2 continued*

pink arrows), a tubule extending and retracting along the same trajectory (ii, green arrows), a tubule bending and moving centripetally between actin filaments (iii, cyan arrows) and an immobile tubule in an actin-poor region (iii, yellow arrows). (**C**) Kymograph analysis of the cell from **A** along the line shown (left) demonstrating coordinated centripetal movement of the ER and actin (see *Video 3*). Time after initial image acquisition is indicated in min: sec; scale bar, 5 µm.

The following figure supplement is available for figure 2:

**Figure supplement 1.** The ER extends at the tips of dynamic microtubules that move radially toward the lamella/lamellipod border.

*Figure 2B*, pink arrows) and occasionally traversed the lamella/lamellipod border, then either rapidly retracted along a similar trajectory (10 of 20 cells; Figure 2Bii, green arrows), or appeared to bend before moving centripetally (20 of 20 cells; Figure 2Biii, cyan arrows). A subset of tubules that penetrated the lamellipod remained relatively immobile in actin-sparse regions (8 of 20 cells; Figure 2Biii, yellow arrows).

What mechanisms determine ER dynamics at the synapse? In general, ER tubules can extend by sliding along the sides of microtubules or by attaching to the tips of growing microtubules (*Waterman-Storer and Salmon, 1998*) through an interaction between STIM1 and the microtubule tip attachment proteins EB1 and EB3 (*Grigoriev et al., 2008*). In Jurkat cells expressing EB1 labeled with EGFP (EB1-EGFP) and ER-mCh, EB1 was seen at the tips of many extending ER tubules (*Figure 2—figure supplement 1A* and *Video 4*), confirming that ER tubules can extend toward the synapse periphery by attaching to the tips of growing microtubules. ER tubules rarely extended into the lamellipod, likely reflecting infrequent microtubule forays into the lamellipod (*Figure 2—figure supplement 1B* and *Video 5*), as has been reported for migrating epithelial cells (*Waterman-Storer and Salmon, 1997*). Like ER tubules, EB1 moved roughly radially though the ADZ and lamella, but at the lamellipod/lamella border the majority reoriented and moved parallel to the synapse edge or disappeared as they moved above the TIRF evanescent field (*Figure 2—figure supplement 1B* and *Video 5*). These findings suggest that ER tubules infrequently enter the lamellipod because microtubules cannot easily penetrate this thin, actin-dense compartment.

Whereas ER tubule elongation was closely associated with microtubule extension, retrograde ER movement was linked to centripetal actin flow. ER tubules undergoing retrograde movement in the lamella were commonly sandwiched between actin arcs (17 of 20 cells; Figure 2Biii, cyan arrows, and *Video 3*), and both moved at the same velocity (*Figure 2C*). Moreover, the similar retrograde velocities of isolated STIM1/Orai1 puncta (47 ± 3 nm/s, mean ± SEM, n = 24 particles; *Figure 1C*) and lamellar actin (36 ± 6 nm/s, mean ± SEM, n = 18 cells; Figure 6F) suggest that retrograde actin flow also sweeps ER-PM junctions into the ADZ. Based on these results, we conclude that the ER at the synapse is dynamic, with peripheral extension on microtubules continually balanced by retraction imposed by retrograde actin flow.

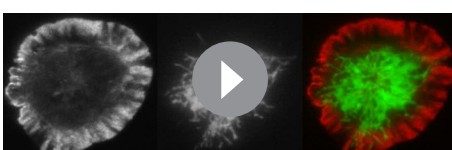

**Video 3.** ER tubules extend from the ADZ into the lamella and are moved centripetally by actin. Time-lapse TIRF movie of a Jurkat cell expressing GFP-actin (left) and ER-mCh (middle) stimulated on an anti-CD3-coated coverslip. A merge of the actin (red) and ER (green) channels is shown at right. Images acquired every 5 s and time compressed 35x. Scale bar, 5 µm. This video supplements *Figure 2*.

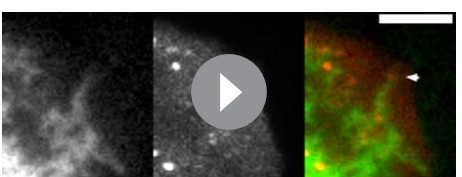

**Video 4.** ER tubules extend toward the synapse edge on the tips of microtubules. Time-lapse TIRF movie of a Jurkat cell expressing ER-mCh (left) and EB1-EGFP (middle) stimulated on an anti-CD3-coated coverslip. A merge of the ER (green) and EB1 (red) channels is shown at right. Images acquired every 5 s and time compressed 35x. Scale bar, 5 µm. This video corresponds to the boxed region in *Figure 2—figure supplement 1A*.

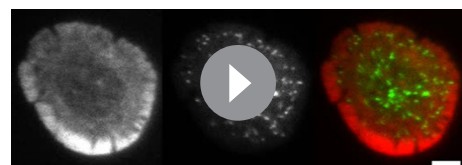

**Video 5.** EB1 moves radially in the ADZ but parallel to the cell edge at the lamella/lamellipod border. Time-lapse TIRF movie of a Jurkat cell expressing F-tractin-P-tdTom (left) and EB1-EGFP (middle) stimulated on an anti-CD3-coated coverslip. A merge of the actin (red) and EB1 (green) channels is shown at right. Images acquired every 1 s and time compressed 7x. Scale bar, 5 μm. This video corresponds to *Figure 2—figure supplement 1B*.

## Calcium influx affects actin organization and dynamics at the synapse

Given the high density of $Ca^{2+}$ influx sites in the ADZ and the known ability of $Ca^{2+}$ to regulate actin dynamics in many cells (*Janmey, 1994*), we asked whether $Ca^{2+}$ influx might acutely regulate retrograde actin flow at the synapse. To this end, we expressed in Jurkat cells the low affinity F-actin binding domain of inositol trisphosphate 3-kinase A labeled with a tandem dimer of fluorescent Tomato (F-tractin-P-tdTom), which allows visualization of filamentous actin without alteration of actin dynamics or function (*Johnson and Schell, 2009*; *Yi et al., 2012*). In the presence of $Ca^{2+}_o$, cells formed a clearly defined lamellipod and lamella (*Figure 3A*, left) with extensive ruffling of the lamellipod and continuous bulk retrograde actin flow as described above in cells expressing GFP-actin (*Video 2*).

Perfusion with $Ca^{2+}$-free medium to terminate $Ca^{2+}$ influx through CRAC channels caused several rapid and profound changes in the organization and dynamics of F-tractin-P at the synapse. In the majority of cells (38 of 45), the distinguishing features of the lamellipod and lamella were lost: ruffling at the periphery was greatly reduced and lamella actin arcs became less apparent, the lamella/lamellipod boundary disappeared as F-actin became more uniformly distributed across the synapse, and in some cells (15 of 45) F-actin extended into the ADZ (*Figure 3A* and *Video 6*). Most strikingly, the centripetal movement of actin was severely reduced and any remaining movement was less radial and more randomly oriented (*Figure 3B* and *Video 6*). These effects all reversed within seconds of restoring $Ca^{2+}_o$, and were also observed in cells expressing GFP-actin (data not shown). Pharmacological inhibition of CRAC channel function with 2-aminoethyldiphenyl borate (2-APB) in the presence of $Ca^{2+}_o$ produced a similar response (11 of 14 cells; *Figure 3C,D*), indicating that changes in actin organization and dynamics result from the inhibition of $Ca^{2+}$ influx through CRAC channels rather than from the removal of $Ca^{2+}_o$ itself.

A critical question is whether these effects of $Ca^{2+}$ on actin in Jurkat leukemic T cells extend to primary human T cells making synapses in a more physiological setting. A recent report has described WASp-associated actin foci in primary T cell synapses that were not detectable in Jurkat cells, suggesting that actin organization in primary T cells may be more complex than previously recognized (*Kumari et al., 2015*). Jurkat cells also lack the lipid phosphatase PTEN, which may affect actin dynamics by enhancing $PIP_3$ accumulation in the plasma membrane (*Shan et al., 2000*). Finally, primary T cells are normally activated by APCs displaying the integrin ICAM-1, which modestly alters actin organization and slows retrograde actin flow at the synapse (*Comrie et al., 2015*). To investigate effects of $Ca^{2+}$ on actin dynamics in a more physiological model, we transduced primary human $CD4^+$ T lymphoblasts with Lifeact-GFP, a short F-actin binding peptide from Abp140 tethered to GFP (*Riedl et al., 2008*). T lymphoblasts plated on anti-CD3- and ICAM-1-coated coverslips formed a distinct lamellipod with rapid retrograde actin flow as described previously (*Comrie et al., 2015*) and $Ca^{2+}$ removal altered the distribution of synaptic actin and reduced ruffling and retrograde flow (20 of 21 cells; *Figure 3E*, *Video 7*). Kymograph analysis showed that on average, $Ca^{2+}$ removal slowed actin flow by 44% and $Ca^{2+}_o$ reperfusion restored it to 97% of the initial velocity (*Figure 3F*, *Table 1*). $Ca^{2+}_o$ removal narrowed the lamellipod by 61%, while $Ca^{2+}_o$ restoration returned the lamellipod to 82% of its initial width (*Figure 3E*, *Table 1*). T lymphoblasts on coverslips coated with anti-CD3 alone produced similar responses (*Video 8*, 40 of 45 cells, *Table 1*). Thus, primary T lymphoblasts and Jurkat T cells responded similarly to changes in $Ca^{2+}_o$, supporting the use of Jurkat T cells as a physiologically relevant model system for studying the effects of $Ca^{2+}$ on actin dynamics at the synapse.

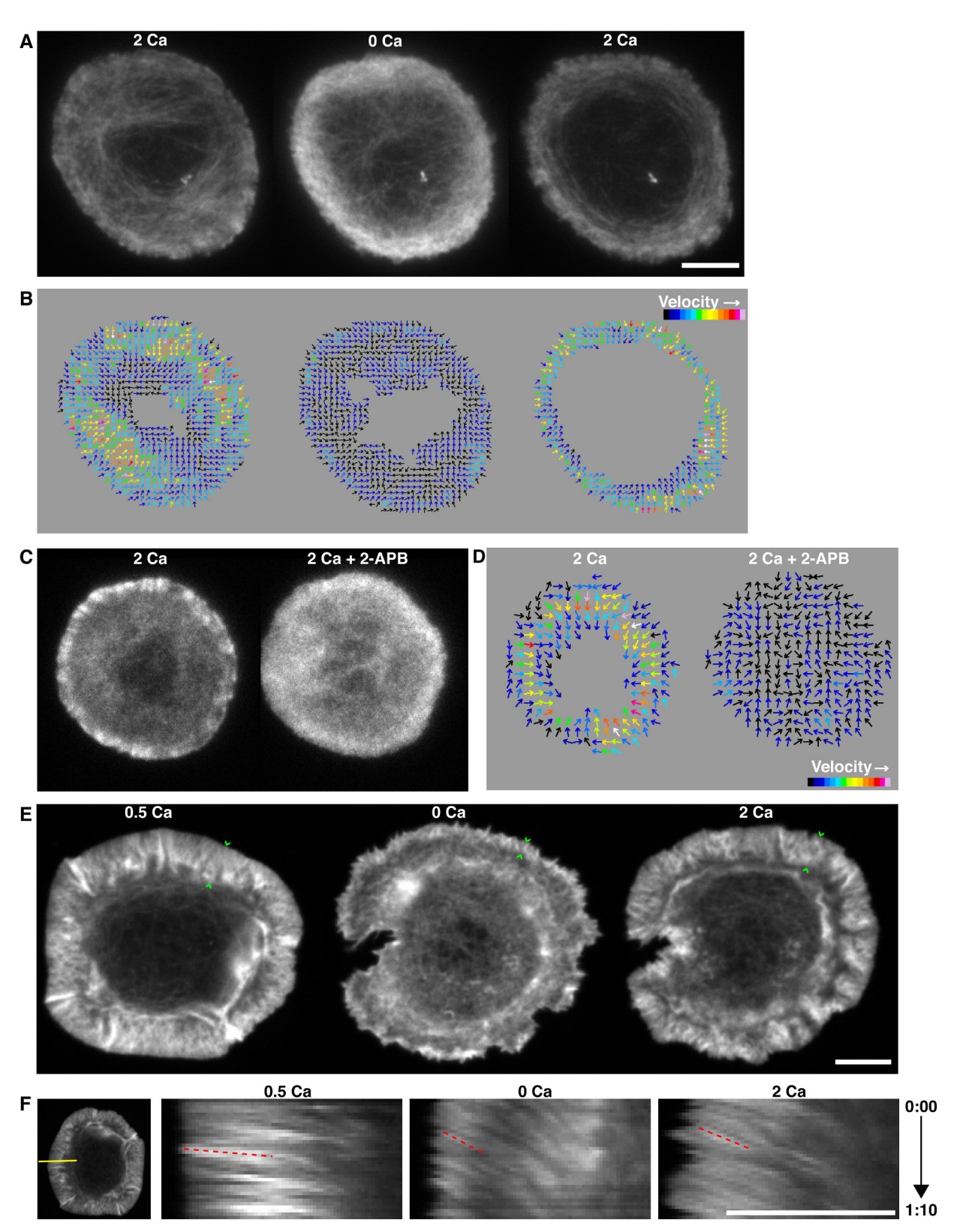

**Figure 3.** Calcium influx organizes synaptic actin and promotes retrograde flow. (**A**) TIRF images of a Jurkat cell expressing F-tractin-P-tdTom after spreading on anti-CD3 in 2 mM $Ca^{2+}_o$ (left), 3.25 min after $Ca^{2+}_o$ removal (center), and 1 min after readdition of 2 mM $Ca^{2+}_o$ (right). $Ca^{2+}$ alters F-actin

*Figure 3 continued on next page*

*Figure 3 continued*

organization and density. Images taken from *Video 6.* (B) Spatiotemporal image correlation spectroscopy (STICS) analysis (*Hebert et al., 2005*) of the cell in **A**, depicting the direction and relative velocity of actin movement before (left) and after $Ca^{2+}_o$ removal (center) and after readdition of 2 mM $Ca^{2+}_o$ (right). Color scale represents relative velocities; numerical values were not assigned because small immobile features cause underestimation of velocity by STICS. (C, D) Blocking $Ca^{2+}$ influx with 2-APB has the same effect on actin as removal of $Ca^{2+}_o$. A representative cell is shown before and 2.5 min after treatment with 100 μM 2-APB, and STICS analysis is shown in **D**. (E) Spinning disk confocal images of a primary human T lymphoblast expressing Lifeact-GFP after spreading on anti-CD3 and ICAM-1 in 0.5 mM $Ca^{2+}_o$ (left), 3 min after $Ca^{2+}_o$ removal (center), and 1.5 min after readdition of 2 mM $Ca^{2+}_o$ (right). The width of the lamellipod (indicated by the green carets) was reduced in 0 $Ca^{2+}_o$. Images are maximum intensity projections of 3 successive 0.25 μm sections of the cell footprint taken from *Video 7.* (F) Kymograph analysis of the cell in **E** along the indicated yellow line (left) demonstrates centripetal actin flow rate of 426 nm/s in 0.5 mM $Ca^{2+}_o$ (left, velocity calculated from the slope of the red dashed lines) that slows to 94 nm/s upon $Ca^{2+}_o$ removal (center) and accelerates to 130 nm/s following readdition of 2 mM $Ca^{2+}_o$. Time is indicated in min:sec; scale bars, 5 μm.

## Calcium influx promotes ER corralling towards the center of the synapse

Because ER tubule movement is influenced by centripetal actin flow (*Figure 2C*), we examined the effect of $Ca^{2+}_o$ removal on ER tubule distribution and dynamics at the Jurkat cell synapse. In the absence of $Ca^{2+}_o$, very few ER tubules were visible in the lamella/lamellipod region by TIRF although small segments of ER were seen near the cell edge, suggesting that under these conditions, peripheral ER tubules extend in the Z dimension out of the TIRF evanescent field (*Figure 4A,B*). Readdition of $Ca^{2+}_o$ initiated a rapid increase in the density of ER tubules in the lamella consistent with their reentry into the evanescent field, and tubules moved centripetally as retrograde actin flow resumed (6 of 6 cells; *Figure 4B,C*, and *Video 9*). These results demonstrate that $Ca^{2+}$ influx through CRAC channels helps to corral extended peripheral ER tubules back to the center of the synapse by promoting retrograde actin flow.

## Calcium reduces the density of F-actin at the synapse

One clue to the mechanism of calcium's effects on actin dynamics at the synapse was that blocking $Ca^{2+}$ influx increased F-tractin-P fluorescence intensity (and thus F-actin density) by 20–30% (*Figure 5A*, *Figure 5—figure supplement 1A*). Elevated F-tractin-P fluorescence did not appear to be a consequence of bulk movement of cellular structures into the evanescent field (such as might result from changes in cell shape) because $Ca^{2+}_o$ removal also increased the F-tractin-P fluorescence at the synapse when viewed by spinning disk confocal microscopy, which

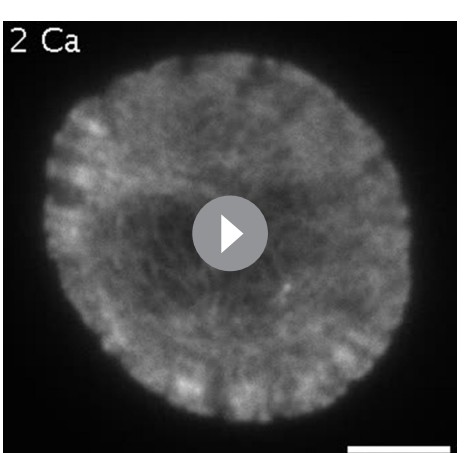

**Video 6.** Calcium influx organizes synaptic actin and promotes retrograde flow in Jurkat cells. Time-lapse TIRF movie of a Jurkat cell expressing F-tractin-P-tdTom after spreading on an anti-CD3 coverslip in 2 mM $Ca^{2+}_o$, followed by perfusion with 0 $Ca^{2+}_o$ and 2 mM $Ca^{2+}_o$. Images acquired every 5 s and time compressed 35x; scale bar, 5 μm. This video supplements *Figure 3A and B*.

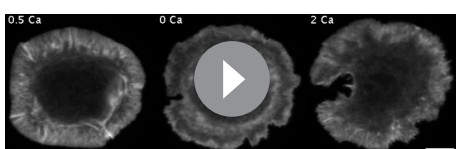

**Video 7.** Calcium influx organizes synaptic actin and promotes retrograde flow in primary human T lymphoblasts plated on anti-CD3 and ICAM-1. Time-lapse spinning disk confocal movie of a human T lymphoblast expressing Lifeact-GFP after spreading on anti-CD3 and ICAM-1 Fc in 0.5 mM $Ca^{2+}_o$ (left), 1.3 min following perfusion with 0 $Ca^{2+}_o$ (center) and 2.2 min after perfusion with 2 mM $Ca^{2+}_o$ (right). Images are displayed as maximum intensity projections of 3 image planes separated by 0.25 μm that were acquired at 2 s intervals. Time compressed 40x; scale bar, 5 μm. This video supplements *Figure 3E and F*.

**Table 1.** Effects of calcium on actin dynamics at the primary T cell immune synapse

| | Anti-CD3 | | | Anti-CD3 + ICAM-1 | | |
|---|---|---|---|---|---|---|
| | 0.5 Ca$^{2+}$ | 0 Ca$^{2+}$ | 2 Ca$^{2+}$ | 0.5 Ca$^{2+}$ | 0 Ca$^{2+}$ | 2 Ca$^{2+}$ |
| Velocity (nm/s) | 243 ± 8 (9) | 145 ± 6 (9) | 208 ± 8 (9) | 167 ± 4 (11) | 94 ± 3 (11) | 162 ± 5 (11) |
| Lamellipod width (μm) | 3.0 ± 0.2 (9) | 1.5 ± 0.1 (9) | 2.3 ± 0.2 (9) | 2.8 ± 0.1 (10) | 1.1 ± 0.1 (10) | 2.3 ± 0.2 (8) |

$[Ca^{2+}]_o$ indicated in mM in the order in which the solutions were applied (see text). Means ± SEM; number of cells indicated in parentheses. Velocities are from a total of 104-133 measurements from kymographs made at 3 different locations per cell.

samples a much thicker optical section (*Figure 5—figure supplement 1B,C*).

To better understand the relationship between F-actin density and $[Ca^{2+}]_i$, we made measurements from single cells expressing F-tractin-P-tdTom and loaded with the $Ca^{2+}$-sensitive dye fura-2. Cells spreading on anti-CD3 in the presence of $Ca^{2+}_o$ had variable $[Ca^{2+}]_i$ consistent with known cell-to-cell variation in proximal TCR signaling in Jurkat cells (*Lewis and Cahalan, 1989*). When both $[Ca^{2+}]_i$ and F-tractin-P fluorescence reached steady-state, $Ca^{2+}_o$ was removed. In some cells stimulated in 2 mM $Ca^{2+}_o$, $[Ca^{2+}]_i$ declined slowly following $Ca^{2+}_o$ removal and did not reach a plateau, probably due to the slow release of mitochondrial $Ca^{2+}$ into the cytosol (*Hoth et al., 1997*). To avoid this complication, we studied synapses formed in 0.5 mM $Ca^{2+}_o$, for which $Ca^{2+}_o$ removal or 2-APB application evoked a rapid and monotonic $[Ca^{2+}]_i$ decline to a similar plateau level in all cells (fura-2 ratio of 0.39 ± 0.04, mean ± SEM, n = 26 cells). The decline in $[Ca^{2+}]_i$ was closely followed by an increase in F-tractin-P fluorescence that plateaued ~30 s after $[Ca^{2+}]_i$ (*Figure 5B*, right). Similarly, readdition of $Ca^{2+}_o$ caused $[Ca^{2+}]_i$ to rise and F-actin to decline, demonstrating a rapidly reversible effect on F-actin density.

Quantifying the relationship between $[Ca^{2+}]_i$ and F-actin concentration is complicated by variation in the expression of F-tractin-P among cells. We therefore quantified the change in F-tractin-P fluorescence in each cell relative to the value in 0 $Ca^{2+}_o$ or following 2-APB application, which produced a similar minimum $[Ca^{2+}]_i$ in all cells. F-tractin-P fluorescence and $[Ca^{2+}]_i$ were measured during the initial response to the TCR stimulus (in 0.5 mM $Ca^{2+}_o$) and following readdition of 2, 5, or 10 mM $Ca^{2+}_o$ (*Figure 5B,C*). In this group of 26 cells the level of F-actin declined as $[Ca^{2+}]_i$ increased, and this relationship was similar regardless of whether measurements were made before or after $Ca^{2+}$ removal. The level of F-actin was highly correlated with $[Ca^{2+}]_i$ ($R^2$ = 0.83) but not with $[Ca^{2+}]_o$ ($R^2$ = 0.22), and 2-APB application and $Ca^{2+}_o$ removal had similar effects on F-tractin-P density, demonstrating that intracellular $Ca^{2+}$ reversibly regulates the density of F-actin at the synapse.

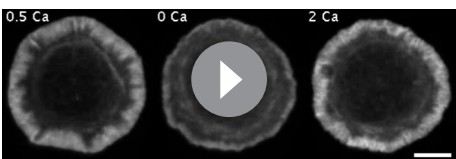

**Video 8.** Calcium influx organizes synaptic actin and promotes retrograde actin flow in primary human T lymphoblasts plated on anti-CD3 without ICAM-1. Time-lapse spinning disk confocal movie of a primary human T lymphoblast expressing Lifeact-GFP after spreading on anti-CD3 in 0.5 mM $Ca^{2+}_o$ (left), 4.3 min following perfusion with 0 $Ca^{2+}_o$ (center) and 2 min following perfusion with 2 mM $Ca^{2+}_o$ (right). Images are displayed as maximum intensity projections of sets of 3 image planes separated by 0.25 μm that were acquired at 2 s intervals. Time compressed 40x; scale bar, 5 μm. This video supplements *Figure 3*.

## Calcium increases the rate of actin depolymerization at the synapse

Given that the steady-state level of F-actin in cells reflects a balance between the overall rates of actin polymerization and depolymerization, $Ca^{2+}$ could reduce the density of F-actin by increasing the rate of depolymerization and/or by reducing the rate of polymerization. We first examined the effect of $Ca^{2+}$ on synaptic actin depolymerization in cells coexpressing F-tractin-P-tdTom and photoactivatable GFP-labeled actin (PAGFP-actin). After brief photoactivation of a small region, the GFP fluorescence indicates only F-actin, because monomeric PAGFP-actin rapidly escapes the region by diffusion; thus, the subsequent decay of fluorescence provides a measure of the actin depolymerization rate (*McGrath et al., 1998*). 3–7 min after settling onto stimulatory coverslips, images of F-tractin-

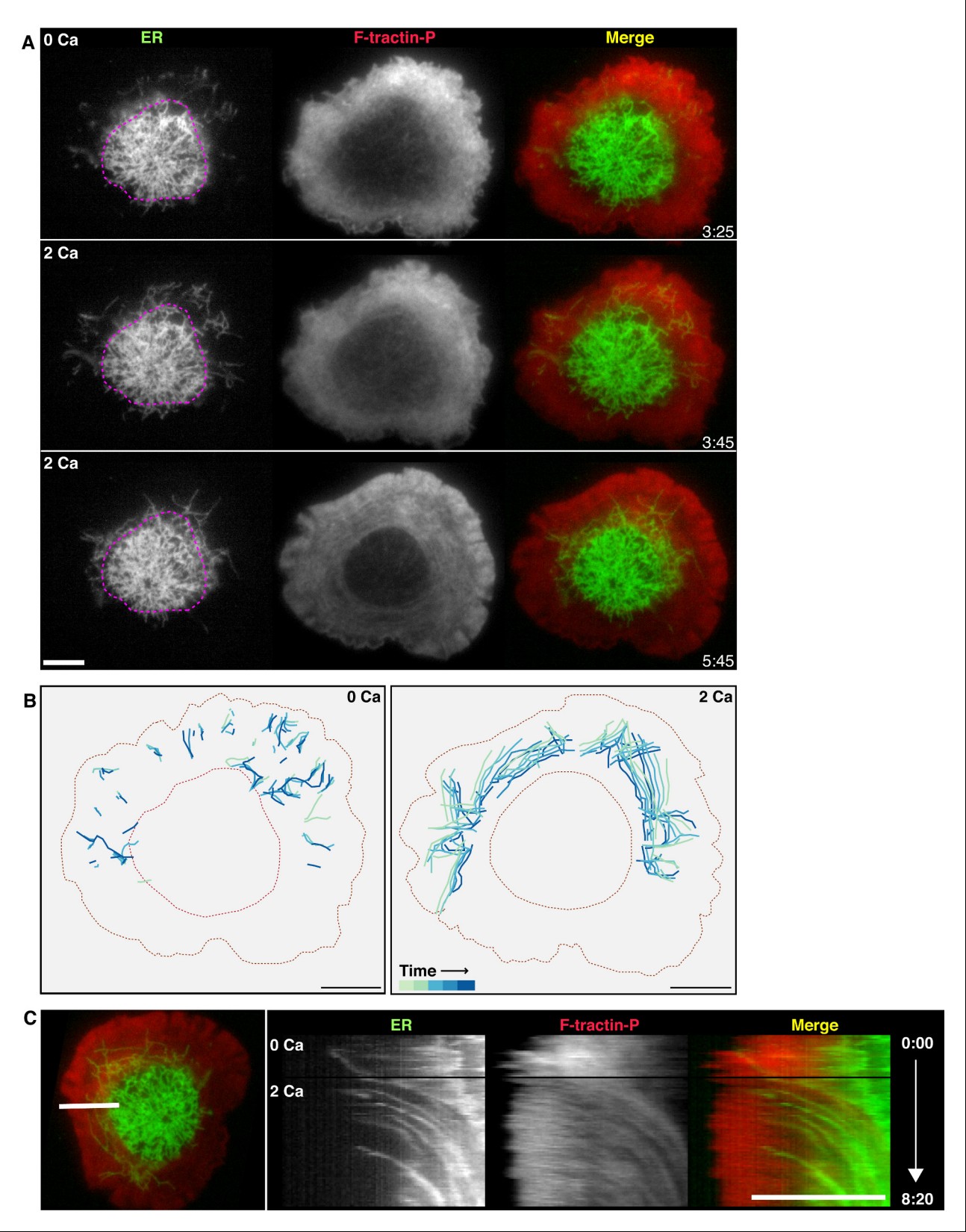

**Figure 4.** Calcium influx promotes ER corralling. (**A**) A cell expressing F-tractin-P-tdTom (red) and ER-GFP (green) on an anti-CD3 coverslip is shown in 0 Ca$^{2+}_o$ and after readdition of 2 mM Ca$^{2+}_o$. Peripheral ER tubules in the TIRF images are sparse in 0 Ca$^{2+}_o$, but Ca$^{2+}_o$ readdition causes peripheral

*Figure 4 continued on next page*

*Figure 4 continued*

tubules to appear as they move into the evanescent field. Pink dotted lines outlining the edge of the ADZ in 0 $Ca^{2+}_o$ serve as a landmark to highlight centripetal ER movement following $Ca^{2+}_o$ readdition. Images taken from *Video 9*. (B) Peripheral ER tubules were traced in 5 images acquired at 10 s intervals, then color-coded for time and overlaid to indicate movement between frames in 0 $Ca^{2+}_o$ (top) and immediately following re-addition of 2 mM $Ca^{2+}_o$ (bottom). In 0 $Ca^{2+}_o$, peripheral tubules are sparse, extended and move in radial and non-radial directions. Peripheral tubules appearing upon readdition of $Ca^{2+}_o$ move centripetally. (C) Kymograph analysis of the cell in A along the indicated line (left) demonstrates centripetal movement of ER tubules between actin structures upon $Ca^{2+}_o$ readdition. Black horizontal lines indicate bath exchange. Gamma adjusted to 0.7 to highlight ER tubules. Time after initial image acquisition is indicated in min:sec; scale bar, 5 μm.

P-tdTom were used to identify cells with steady-state treadmilling actin and small (~1 by 3 μm) regions in the lamella and lamellipod were defined for photoactivation (*Figure 6A*, left, red ovals). PAGFP actin was first photoactivated in the presence $Ca^{2+}_o$ and again ~2 min following $Ca^{2+}_o$ removal when F-actin had reached a new steady-state (*Figure 6A*, *Figure 6—figure supplement 1A–D* and *Video 10*). After each photoactivation, the resulting GFP fluorescence was measured over time by widefield microscopy rather than TIRF in order to prevent any change in signal due to possible movement of actin in the Z direction, and translocation of the photoactivated region was used to measure centripetal velocity.

With $Ca^{2+}_o$ present, photoactivated actin moved toward the center of the synapse at velocities comparable to those previously reported from kymograph measurements (*Yi et al., 2012*; *Babich et al., 2012*), and moved faster in the lamellipod than in the lamella as expected (*Figure 6B, C, F* and *Table 2*). In the absence of $Ca^{2+}_o$, mean actin velocity decreased by 41% in the region of the lamellipod and by 64% in the lamella (*Figure 6B, C, F* and *Table 2*).

We measured the rate of actin filament depolymerization from the single exponential decay of GFP fluorescence with time (*Figure 6D,E* and *Figure 6—figure supplement 1A–D*). The decay kinetics in 2 mM $Ca^{2+}_o$ were similar in the lamella and lamellipod, and $Ca^{2+}$ removal extended the mean actin filament half-life by 17% in the lamella and 23% in the lamellipod region (*Figure 6G* and *Table 2*). To control for possible effects of shear force during perfusion or $Ca^{2+}$-independent changes in actin dynamics over time, we photoactivated regions before and after perfusing the cell with a solution containing the same $[Ca^{2+}]$ and found no significant change in either velocity or half-life (*Figure 6—figure supplement 1E,F*). Thus, our findings indicate that $Ca^{2+}$ influx accelerates actin depolymerization at the synapse.

$Ca^{2+}$ can enhance actin depolymerization through many effectors. Myosin IIA seemed a likely candidate because it is present at the synapse (*Ilani et al., 2007*; *Jacobelli et al., 2004*; *Babich et al., 2012*; *Yi et al., 2012*), is known to disassemble actin filaments in the lamella of epithelial cells (*Wilson et al., 2010*), and can be activated by $Ca^{2+}$ (*Kamm and Stull, 1985*). However, after inhibition of myosin ATPase activity with blebbistatin, $Ca^{2+}_o$ removal induced a 30% increase in F-actin density (*Figure 6—figure supplement 2A,B*), similar to its effect in the absence of the drug (*Figure 5C*). This was not due to a failure to inhibit myosin ATPase activity because blebbistatin treatment caused actin arcs to accumulate in the ADZ as previously reported (*Yi et al., 2012*). These results suggest that $Ca^{2+}$ regulates F-actin density and depolymerization at the synapse independently of myosin.

## Calcium restricts actin polymerization to the distal edge of the lamellipod

A second potential mechanism by which $Ca^{2+}$ could alter F-actin density at the synapse is by influencing actin polymerization. To test for such an effect, we photoactivated PAGFP-actin to release a pool of fluorescent actin monomers and monitored their incorporation into actin filaments. Cells expressing F-tractin-P-tdTom and

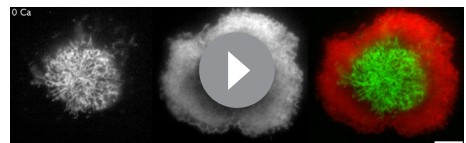

**Video 9.** Calcium-dependent retrograde actin flow corrals the ER in the ADZ. Time-lapse TIRF movie of a Jurkat cell expressing ER-GFP (left) and F-tractin-P-tdTom (center) that had spread in 2 mM $Ca^{2+}_o$ before perfusion with 0 $Ca^{2+}_o$. Video begins with the cell in 0 $Ca^{2+}_o$ and shows the effect of restoring 2 mM $Ca^{2+}_o$. A merge of the ER (green) and F-tractin-P (red) channels is shown at right. Images acquired every 5 s and time compressed 35x. Scale bar, 5 μm. This video supplements *Figure 4A–C*.

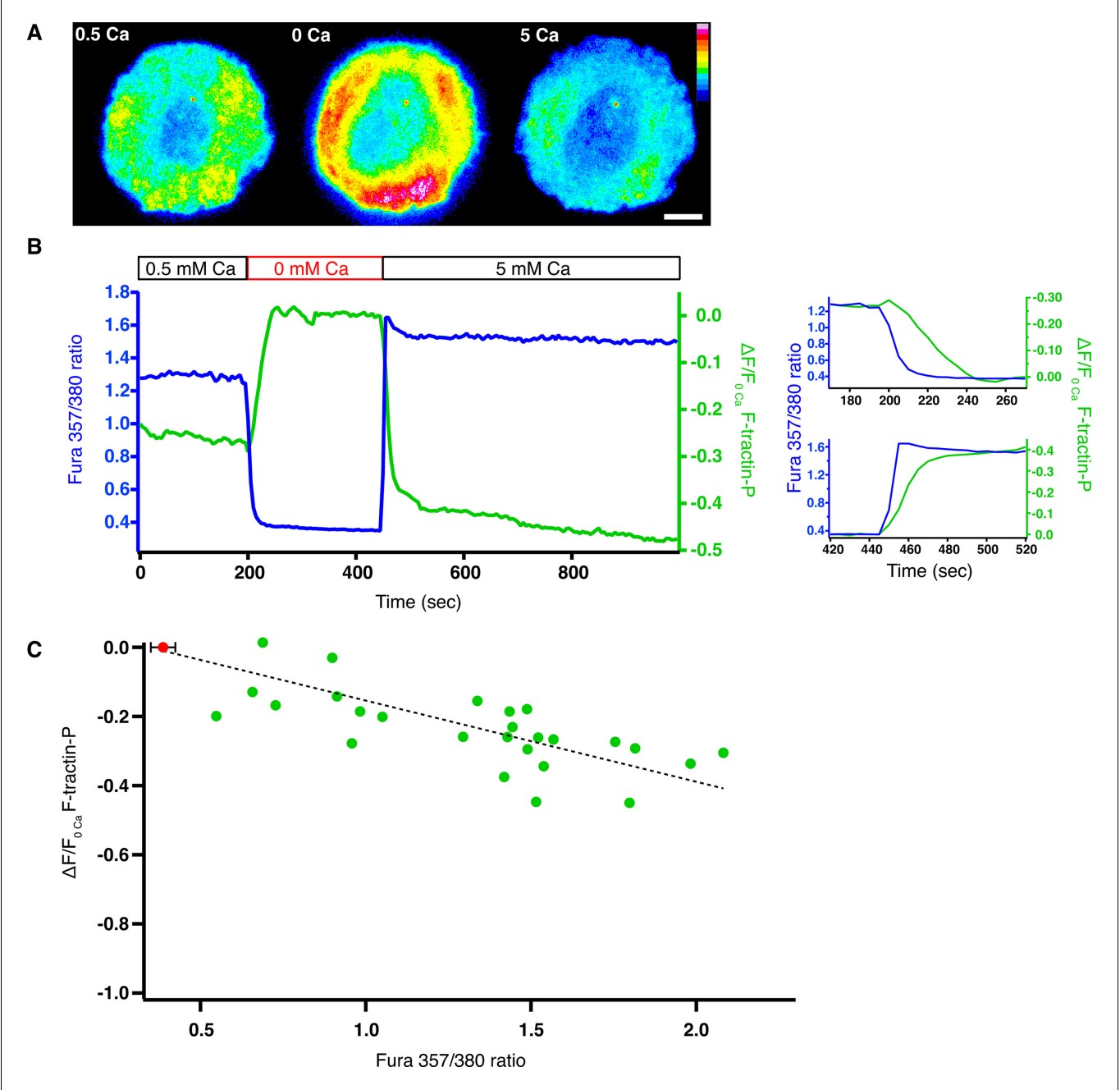

**Figure 5.** Intracellular calcium reduces the density of F-actin at the synapse. Jurkat T cells expressing F-tractin-P-tdTom and loaded with fura-2 were stimulated on anti-CD3-coated coverslips. (**A**) Pseudocolor image of F-tractin-P-tdTom intensity in a cell exposed sequentially to 0.5 mM $Ca^{2+}_o$ (left), 0 $Ca^{2+}_o$ (center) and 5 mM $Ca^{2+}_o$ (right), indicating a $Ca^{2+}$-dependent decrease in F-actin density. Linear color scale indicates fluorescence intensity (0–1 a.u.); scale bar, 5 μm. (**B**) Change in F-tractin-P-tdTom fluorescence (green; relative to fluorescence in 0 $Ca^{2+}_o$) and fura-2 ratio (blue) from the cell in **A**. The data are replotted on the right with an inverted F-tractin-P axis to highlight the delay between changes in $[Ca^{2+}]_i$ and F-tractin-P intensity upon $Ca^{2+}_o$ removal (top)and readdition (bottom). (**C**) Change in F-tractin-P-tdTom fluorescence (relative to fluorescence in 0 $Ca^{2+}_o$ or 100 μM 2-APB in 0.5 mM $Ca^{2+}_o$) as a function of fura-2 ratio. Each point is an average single-cell value measured at constant fura-2 ratio and F-tractin-P fluorescence in the presence of 0.5–10 mM $Ca^{2+}_o$ (green). The red dot indicates the average baseline fura-2 ratio ( ± s.d.) for all cells in 0 $Ca^{2+}_o$ or 2-APB. A linear fit to the data is shown ($r^2$ = 0.83).

The following figure supplement is available for figure 5:

**Figure supplement 1.** $Ca^{2+}$ influx through CRAC channels reduces F-actin density at the synapse.

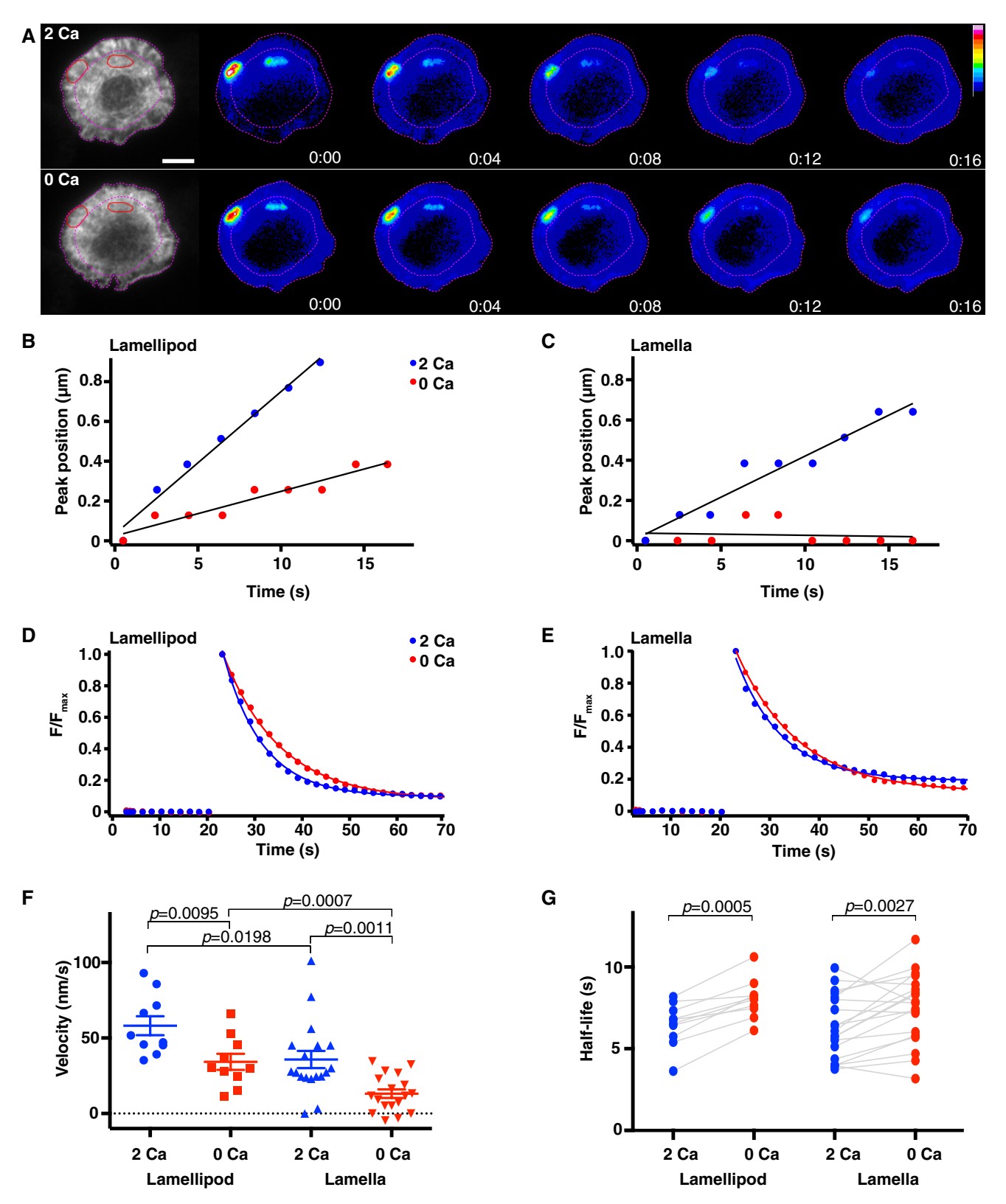

**Figure 6.** Calcium accelerates actin depolymerization and centripetal velocity at the synapse. Jurkat T cells expressing F-tractin-P-tdTom and PAGFP-actin were stimulated on anti-CD3 coated coverslips and TIRF images of F-tractin-P-tdTom (A, left) were used to identify regions in the lamellipod and *Figure 6 continued on next page*

*Figure 6 continued*

lamella (red ovals) to photoactivate. The lamella/lamellipod border in 2 mM $Ca^{2+}_o$ and cell edge are indicated by pink dashed lines. (**A**) Widefield epifluorescence images of PAGFP-actin after photoactivation in 2 mM $Ca^{2+}_o$ (top) and a subsequent photoactivation in 0 $Ca^{2+}_o$ (bottom). Images are from *Video 10*; color scale indicates fluorescence intensity (0–1 a.u.). Time after photoactivation indicated in min:sec. Scale bar, 5 µm. (**B, C**) Position of peak PAGFP-actin fluorescence as a function of time after photoactivation in the lamellipod (**B**) and the lamella (**C**) (see figure supplement 1A-D). Data are plotted in the presence (blue) and absence (red) of $Ca^{2+}_o$ for the cell pictured in **A**. Linear fits to the data indicate lamellipod velocities of 72 nm/s (2 Ca) and 22 nm/s (0 Ca) and lamella velocities of 41 nm/s (2 Ca) and 1 nm/s (0 Ca). (**D, E**) The fluorescence decay of photoactivated PAGFP-actin in the lamellipod (**D**) and lamella (**E**) for the cell in **A** was fitted by a single exponential. In the lamellipod, $\tau$ = 8.3 s (2 Ca) and 12.0 s (0 Ca); in the lamella, $\tau$ = 9.2 s (2 Ca) and 12.4 s (0 Ca). F/$F_{max}$ is the fluorescence intensity after photoactivation relative to the peak. (**F**) The centripetal velocity of photoactivated actin in the lamellipod (n = 10 cells) and lamella (n = 18 cells) in the presence of absence of $Ca^{2+}_o$, calculated as described in **B, C**. Error bars indicate SEM; p-values from Student's two-tailed t-test. (**G**) Actin filament half-life calculated from the exponential rate of fluorescence decay in photoactivated regions in the lamellipod (n = 10) and lamella (n = 19) with and without $Ca^{2+}_o$. P-values are from paired Student's two-tailed t-test.

The following figure supplements are available for figure 6:

**Figure supplement 1.** Actin filament velocity and half-life at the synapse.

**Figure supplement 2.** Calcium influx alters actin organization and density independently of myosin activity.

PAGFP-actin were stimulated on anti-CD3 coverslips and F-tractin-P-tdTom images were used to identify cells with steady-state treadmilling actin. PAGFP-actin was photoactivated within the ADZ, where the majority of actin is expected to be monomeric and freely diffusible, in the presence of $Ca^{2+}_o$ or ~1.5 min after its removal, a time when $[Ca^{2+}]_i$ would be expected to reach a constant minimum (see *Figure 5B*; *Video 11*). After each photoactivation, the incorporation of fluorescent PAGFP-actin monomers into filaments throughout the cell was visualized over time by TIRF.

In the absence of $Ca^{2+}_o$, GFP fluorescence increased immediately in the ADZ upon photoactivation, followed by a slower rise throughout the lamella as fluorescent actin monomers diffused through the cytosol and incorporated into F-actin. The fluorescence rise reached the cell's edge within 3 s of photoactivation, consistent with the rapid diffusion of monomeric actin in cells, and covered most of the actin-rich area of the synapse, reflecting widespread polymerization (*Figure 7A,C*). In the presence of $Ca^{2+}_o$, actin polymerization was strikingly different. Within 2 s of photoactivation in the ADZ, fluorescence increased selectively in a narrow band around the periphery of the lamellipod (*Figure 7B*). Fluorescence at the periphery peaked within ~6 s, then declined slightly as the combination of peripheral incorporation and centripetal flow labeled the entire lamellipod, generating a wide band of fluorescent actin that dropped off sharply at the lamella/lamellipod border (*Figure 7B,D*). In contrast, fluorescence in the lamella increased only minimally during the 30 s following photoactivation. The reduced polymerization in the lamella was not due to the inability of PAGFP-actin to incorporate into lamellar structures (*Yi et al., 2012*) because PAGFP-actin incorporated efficiently into the lamella in 0 $Ca^{2+}_o$. These results demonstrate that $Ca^{2+}$ influx effectively promotes actin polymerization at the distal edge of the lamellipod while suppressing polymerization elsewhere throughout the synapse.

One potential mechanism for directing actin polymerization to the lamellipod edge is through selective localization of F-actin nucleation complexes. The nucleation promotion factor WAVE2 is essential for normal F-actin accumulation at the synapse, and a protein complex including WAVE2 and Abi1 has been detected at the synapse periphery (*Zipfel et al., 2006*; *Nolz et al., 2007*), but whether $Ca^{2+}$ influences the localization of this complex is not known. Using immunocytochemistry we found

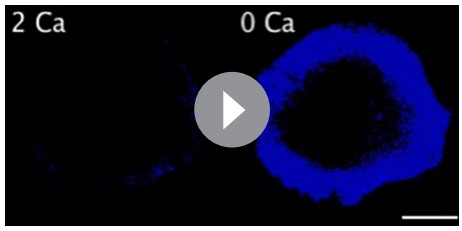

**Video 10.** Calcium increases actin depolymerization and centripetal velocity at the synapse. Time-lapse TIRF movie of a Jurkat cell expressing F-tractin-P-tdTom and PAGFP-actin. Bars of PAGFP-actin are photoactivated in the lamella and lamellipod of the same cell in 2 mM $Ca^{2+}_o$ (left) and 0 $Ca^{2+}_o$ (right). Intensity is rendered in pseudocolor using the scale in *Figure 6A*. Images acquired every 2 s and time compressed 6x. Scale bar, 5 µm. The video supplements *Figure 6A*.

**Table 2.** Effects of calcium on actin dynamics at the Jurkat cell immune synapse

| | Lamellipod | | Lamella | |
|---|---|---|---|---|
| | 2 $Ca^{2+}$ | 0 $Ca^{2+}$ | 2 $Ca^{2+}$ | 0 $Ca^{2+}$ |
| Half-life (s) | 6.5 ± 0.4 (10) | 8.0 ± 0.4 (10) | 6.4 ± 0.4 (19) | 7.5 ± 0.5 (19) |
| Velocity (nm/s) | 58 ± 6 (10) | 34 ± 5 (10) | 36 ± 6 (18) | 13 ± 3 (18) |

$[Ca^{2+}]_o$ indicated in mM. Means ± SEM; number of cells indicated in parentheses.

that $Ca^{2+}_o$ enriched the level of endogenous WAVE2 at the synapse periphery (**Figure 7E,F**). To examine the timing of the WAVE2 complex response to $Ca^{2+}$, we visualized EGFP-Abi1 in live cells. Abi1 was highly enriched at the synapse periphery in the presence of $Ca^{2+}_o$ as expected (**Video 12**, **Figure 7G**, left, right panels), and $Ca^{2+}_o$ removal caused Abi1 to become diffusely distributed throughout the synapse in a reversible manner (**Figure 7G**, middle, **Figure 7H,I**). The finding that $Ca^{2+}$ localizes the WAVE2 complex to the lamellipod edge suggests that $Ca^{2+}$ restricts actin polymerization to the synapse periphery at least in part through controlling the location of nucleation.

## Discussion

Extensive reorganization of the actin cytoskeleton underlies the formation of the immune synapse, and retrograde actin flow from the lamellipod towards the ADZ is critical for maintaining TCR signaling and $[Ca^{2+}]_i$ elevation (**Valitutti et al., 1995**; **Varma et al., 2006**; **Rivas et al., 2004**; **Babich et al., 2012**; **Yi et al., 2012**). While the critical role of $[Ca^{2+}]_i$ elevation in regulating gene expression during T cell activation is well established (**Feske et al., 2001**), the current study reveals several essential new functions for $Ca^{2+}$ in determining synapse form and function. $Ca^{2+}$ influx through CRAC channels organizes actin into distinct lamella and lamellipod zones, stimulates retrograde actin flow and concentrates active CRAC channels in the center of the synapse, in part through its action to return extending ER tubules to the ADZ.

Our findings extend upon previous work implicating $Ca^{2+}$ in actin remodeling at the synapse. In a pioneering study, Bunnell et al demonstrated that intracellular $Ca^{2+}$ is required for actin accumulation and cell spreading during the early phase of synapse formation; however, the failure of the cells to spread in the absence of $Ca^{2+}_o$ precluded study of calcium's effects on the mature synapse (**Bunnell et al., 2001**). Likewise, $Ca^{2+}$-sensitive proteins including L-plastin (**Wabnitz et al., 2010**), gelsolin (**Morley et al., 2007**), calpain (**Watanabe et al., 2013**) and myosin IIA (**Ilani et al., 2009**; **Yi et al., 2012**) have been implicated in actin remodeling at the synapse, but in these studies protein expression levels or activity were perturbed prior to synapse initiation, and thus effects on cell adhesion and spreading could not be distinguished from possible effects on actin dynamics in the mature synapse. We were able to address the role of $Ca^{2+}$ in the mature synapse by acutely blocking $Ca^{2+}$ influx after the synapse was fully formed as indicated by its stable contact area, well-defined lamellipod and lamella actin zones, and retrograde actin flow. In both Jurkat cells and primary T lymphoblasts, $Ca^{2+}_o$ removal triggered similar changes in actin organization and retrograde flow. $Ca^{2+}_o$ removal slowed retrograde flow in the lamellipod to nearly the same degree in both cells (41% decrease in Jurkat versus 44% in primary T cells; **Tables 1** and **2**). $Ca^{2+}$ removal diminished the lamellipod in primary T cells, reducing its width by 61%, while in Jurkat cells the effect was somewhat more pronounced with the lamellipod

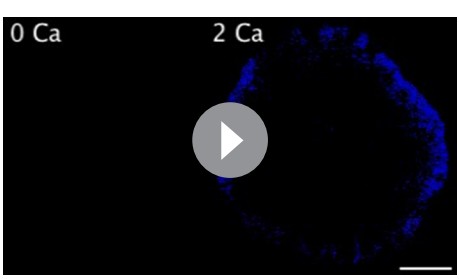

**Video 11.** Calcium restricts actin polymerization to the distal edge of the synapse. Time-lapse TIRF movie of a Jurkat cell expressing F-tractin-P-tdTom and PAGFP-actin. Monomeric PAGFP-actin is photoactivated in the ADZ of two different cells in 2 mM $Ca^{2+}_o$ (left) and 0 $Ca^{2+}_o$ (right). Intensity is rendered in pseudocolor using the scale in **Figure 7A**. Images acquired every 500 ms and time compressed 3.5x. Scale bar, 5 μm. This video supplements **Figure 7A and B**.

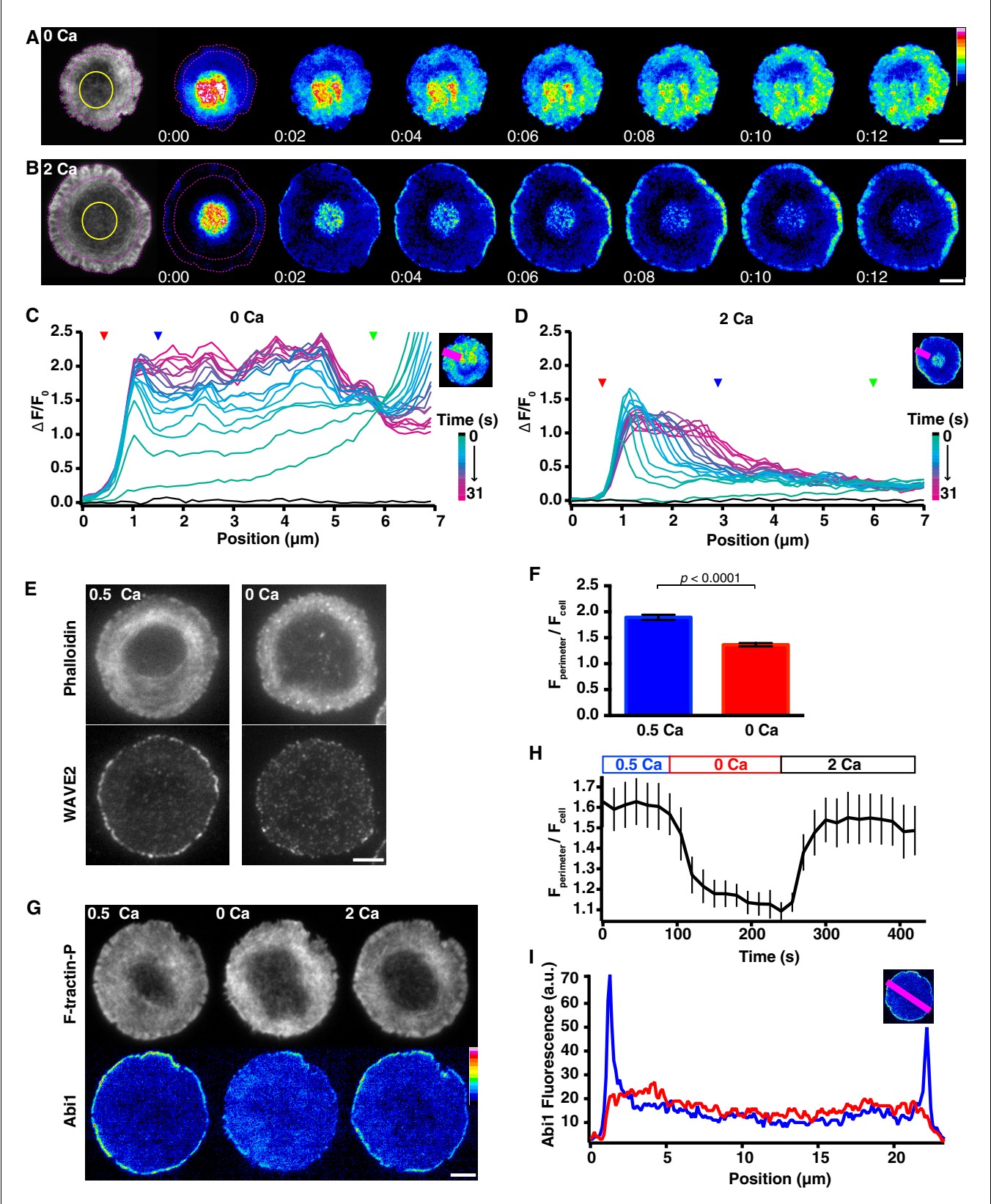

**Figure 7.** Calcium restricts actin polymerization to the distal edge of the synapse. (A, B) Two Jurkat T cells expressing F-tractin-P-tdTom and PAGFP-actin were stimulated on anti-CD3 coated coverslips in 0.5 mM $Ca^{2+}$, and PAGFP-actin was photoactivated in the ADZ regions indicated in the
*Figure 7 continued on next page*

**Figure 7 continued**

F-tractin-P-tdTom TIRF images (left, yellow circles) 2 min after perfusion of 0 $Ca^{2+}_o$ (**A**) or 2 mM $Ca^{2+}_o$ (**B**). Incorporation of fluorescent PAGFP-actin is shown as a function of time after photoactivation. The lamella/lamellipod border in 2/0.5 mM $Ca^{2+}_o$ and cell edge are indicated by pink dashed lines. Images are from *Video 11*. Time after photoactivation is in min:sec; scale bar, 5 μm; color scale indicates fluorescence intensity (0–1 a.u.). (**C, D**) Normalized PAGFP-actin fluorescence intensity (see Materials and methods) along the line indicated (top right) as a function of radial position. The fluorescence profile before photoactivation is shown in black; the color scale applies to subsequent profiles acquired every 1.5 s after photoactivation. The cell edge (red arrowhead), the lamellipod/lamella border (blue arrowhead) and the edge of the ADZ (green arrowhead) are indicated. Data are representative of 12–13 cells. (**E**) Representative TIRF images of Jurkat cells stimulated on anti-CD3 in 0.5 mM $Ca^{2+}$ then transferred to 0.5 $Ca^{2+}$ (left) or 0 $Ca^{2+}$ (right) for 2.5 min, labeled with Alexa-594 phalloidin (top) and anti-WAVE2 (bottom) (see Materials and methods). $Ca^{2+}$ promotes localization of WAVE2 to the edge of the lamellipod. (**F**) Average anti-WAVE2 fluorescence in a 1-μm band around the perimeter of the synapse ($F_{perimeter}$) relative to the average fluorescence across the whole synapse ($F_{cell}$) in 0 and 0.5 mM $Ca^{2+}$ (n = 63 cells each). Error bars indicate SEM; p-values from Student's two-tailed t-test. (**G**) TIRF images of a Jurkat cell expressing F-tractin-P-tdTom (top) and EGFP-Abi1 (bottom) stimulated on anti-CD3 in 0.5 mM $Ca^{2+}_o$ (left), 1.5 min after $Ca^{2+}_o$ removal (center), and 1.5 min after readdition of 2 mM $Ca^{2+}_o$ (right). $Ca^{2+}$ promotes Abi1 localization to the edge of the lamellipod. Images are taken from *Video 12*. Scale bar, 5 μm; color scale indicates fluorescence intensity (0–1 a.u.). (**H**) The average fluorescence of EGFP-Abi1 in a 1-μm band around the perimeter of the synapse ($F_{perimeter}$) relative to the average fluorescence across the entire synapse ($F_{cell}$) versus time (n = 8 cells). (**I**) The fluorescence intensity (a.u.) of EGFP-Abi1 along the line indicated (top right, pink) in 0.5 mM $Ca^{2+}_o$ (blue) and 1.5 min after $Ca^{2+}_o$ removal (red).

becoming indistinguishable from the rest of the actin network. Overall, these results demonstrate that $Ca^{2+}$ effects on actin organization and dynamics are not specific to Jurkat cells but apply also to primary T cells. Furthermore, T lymphoblasts responded similarly to $Ca^{2+}_o$ removal when stimulated on surfaces including ICAM-1 in order to more closely resemble an APC, indicating that our findings extend to more physiological surface interactions. New optical technologies such as light sheet microscopy may enable further studies of $Ca^{2+}$ effects on actin dynamics in the most physiological setting, at the synapse between a primary T cell and an APC (*Ritter et al., 2015*).

We found that $Ca^{2+}$ acts at multiple levels to organize actin into a lamellipod and lamella with sustained retrograde flow. First, $Ca^{2+}$ directs actin polymerization largely to the distal edge of the lamellipod while suppressing it elsewhere (*Figure 7*). By effectively restricting polymerization to the edge, $Ca^{2+}$ suppresses filament growth at random angles throughout the synapse and promotes retrograde vectorial movement of actin filaments (*Figure 8*). Second, $Ca^{2+}$ accelerates actin depolymerization (*Figure 6*), which is expected to enhance the rate of actin flow further by increasing the level of monomeric actin and therefore the rate of actin addition to free barbed ends at the lamellipod edge (*Figure 8*). Overall, these findings demonstrate that $Ca^{2+}$ adds a second level of regulation to actin remodeling downstream of TCR triggering. The net effect of $Ca^{2+}$ influx is to suppress the density of F-actin at the synapse (*Figure 5*), which was somewhat surprising given that elevating intracellular $Ca^{2+}$ has been reported to increase the level of F-actin in unstimulated T cells (*Dushek et al., 2008*). This discrepancy may have resulted from the different imaging methods that were used; flow cytometry may indicate an increased level of F-actin globally (*Dushek et al., 2008*), while TIRF or confocal imaging at the cell footprint may instead detect a local decrease in F-actin at the synapse (this study). Alternatively, TCR triggering may initiate $Ca^{2+}$-sensitive actin regulatory pathways that are quiescent in resting cells, and thus $Ca^{2+}$ engages a different set of actin remodeling proteins after TCR stimulation.

Calcium acts in two ways to spatially restrict actin polymerization at the mature synapse: by promoting polymerization around the cell perimeter and by suppressing polymerization throughout the rest of the contact area. The extensive polymerization at the lamellipod edge is closely paralleled by the recruitment of the WAVE2/Abi1 complex to the periphery (*Figure 7*), where it presumably activates Arp2/3 to initiate actin

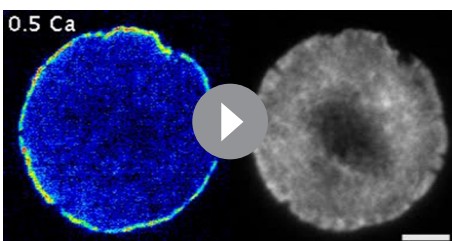

**Video 12.** Calcium promotes localization of Abi1 to the distal edge of the synapse. Time-lapse TIRF movie of a Jurkat cell expressing F-tractin-P-tdTom and EGFP-Abi1 stimulated on anti-CD3 coated coverslip in 0.5 mM $Ca^{2+}_o$, followed by perfusion with 0 $Ca^{2+}_o$ and 2 mM $Ca^{2+}_o$. Images acquired every 15 s and time compressed 45x; scale bar, 5 μm. This video supplements *Figure 7G, H and I*.

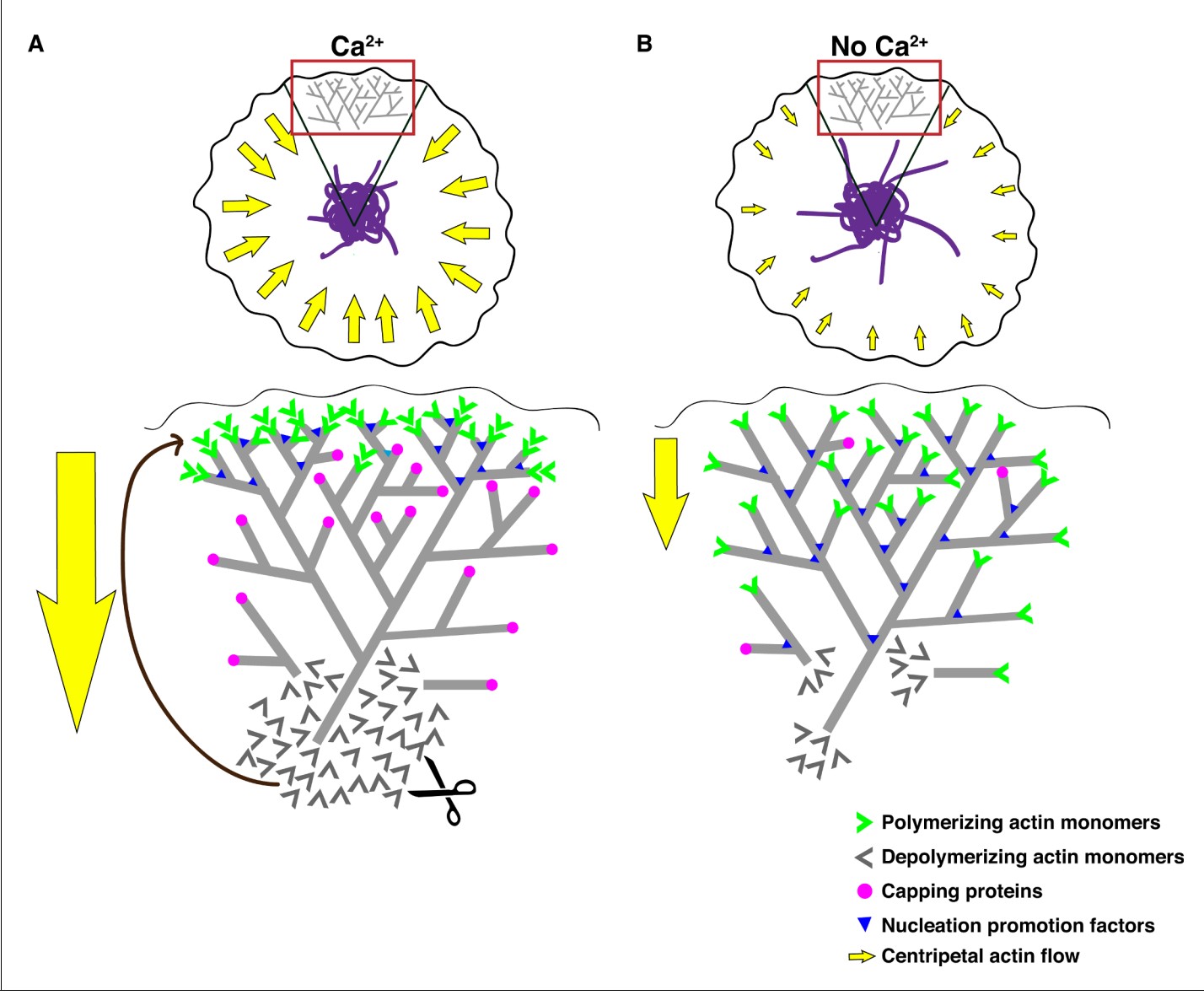

**Figure 8.** Effects of calcium on actin dynamics and retrograde flow at the synapse. (**A**) Retrograde actin flow at the immune synapse (yellow arrows) continually removes extended ER tubules (purple) from the periphery, thereby concentrating the ER in the ADZ. An expanded view of the red-boxed region (top) depicts $Ca^{2+}$ effects on actin regulation (bottom). $Ca^{2+}$ drives centripetal actin flow in two ways: (1) by restricting polymerization to the lamellipod edge (green chevrons), it enforces vectorial movement of the actin network; and (2) by increasing the rate of depolymerization, it increases the pool of free actin monomers (grey chevrons), thus enhancing polymerization on free barbed ends at the lamellipod edge (green chevrons). $Ca^{2+}$ restricts polymerization to the lamellipod edge by localizing WAVE2 and Abi1 to this site where they promote ARP2/3-mediated actin nucleation (blue triangles) and possibly by capping free barbed ends elsewhere (pink circles). (**B**) Experimentally terminating $Ca^{2+}$ influx reduces retrograde actin flow such that extended ER tubules are no longer effectively pushed into the ADZ. In the absence of $Ca^{2+}_{o}$, actin depolymerization is reduced (bottom), nucleation occurs more uniformly throughout the lamellipod/lamella and capping of free barbed ends may be reduced. The overall result is a slowed, non-directional polymerization throughout the lamellipod and lamella resulting in reduced retrograde flow.

polymerization and branching (*Takenawa and Suetsugu, 2007*). A central question arising from these findings is how $Ca^{2+}$ directs WAVE2/Abi1 to the periphery. One possible mechanism is suggested by the ability of $Ca^{2+}$ to stimulate PI3K localization to the lamellipod of migrating cells, where generation of phosphatidylinositol (3,4,5)-trisphosphate (PIP$_3$) may recruit the WAVE2 complex to the membrane (*Oikawa et al., 2004*). $Ca^{2+}$ may exert additional effects through its ability to promote GTP loading of Rac, a GTPase essential for WAVE2-mediated actin nucleation (*Fleming et al.,*

*1999*). The mechanism by which $Ca^{2+}$ suppresses polymerization elsewhere in the lamella and lamel-lipod is also unknown, although $Ca^{2+}$-sensitive capping proteins expressed in T cells such as gelsolin (*Yin, 1987*) or CapG (*Yu et al., 1990*) are attractive candidates. Our results indicate that $Ca^{2+}$-dependent acceleration of depolymerization is unlikely to involve myosin (*Figure 6—figure supplement 2A,B*), but the actin-severing proteins cofilin (*Maus et al., 2013*; *Meberg et al., 1998*; *Wang et al., 2005*) and gelsolin (*Yin, 1987*) remain viable candidates as both are expressed in T cells and respond to physiological levels of $[Ca^{2+}]_i$ (*Lin et al., 2000*).

Because the ER-PM junction forms the physical site for STIM1-Orai1 assembly into active CRAC channels, the location and dynamics of the ER are critical factors that determine when and where $Ca^{2+}$ influx sites arise when T cells contact their targets. Our results provide the first view of ER dynamics at the synapse and how the actin cytoskeleton restricts both the ER and CRAC channel distribution to the cSMAC/ADZ. As the synapse forms, STIM1 and Orai1 appear in the cSMAC/ADZ by two mechanisms. The first is related to the movement of the centrosome and associated MTOC to the synapse, as EM tomography has shown enrichment of the ER around the centrosome at synaptic contact sites (*Ueda et al., 2011*). This mechanism appears to account for the bulk of ER localization and STIM1-Orai1 complexes as cells settled onto coverslips. Once a stable synapse formed, microtubule extension carried ER tubules toward the periphery, and these were repeatedly returned to the ADZ by an advancing front of actin. A similar action of actin to oppose the extension of microtubules and associated ER tubules has also been described at the leading edge of migrating epithelial cells (*Terasaki and Reese, 1994*; *Waterman-Storer and Salmon, 1997*). Our observations that STIM1/Orai1 puncta and actin move towards the ADZ at similar speeds suggest that nascent ER tubules may form ER-PM junctions in peripheral regions, which then enable the assembly of active STIM1-Orai1 complexes that traverse the lamella before they are collected in the ADZ. Interestingly, intra-cellular $Ca^{2+}$ binding to membrane phospholipids is thought to enhance TCR signaling by exposing membrane-associated CD3 ITAM motifs (*Shi et al., 2013*). Thus, an intriguing possibility is that mobile CRAC channel complexes create local sites of high $[Ca^{2+}]_i$ needed to fully activate mobile TCR microclusters.

This study illustrates two new levels of signal regulation at the immune synapse. Because actin dynamics are required to sustain TCR activity at the synapse (*Kaizuka et al., 2007*; *Valitutti et al., 1995*; *Varma et al., 2006*; *Rivas et al., 2004*; *Babich et al., 2012*; *Yi et al., 2012*; *Kumari et al., 2015*), the action of $Ca^{2+}$ influx to promote actin turnover and flow creates a positive feedback loop that would be expected to maintain or enhance the activation of CRAC channels. This loop creates the potential for nonlinear effects, such that graded increases or decreases in $[Ca^{2+}]_i$ may act through actin to modulate TCR activity and bias the cell towards all-or-none, threshold-like behavior in response to antigen. At the same time, $Ca^{2+}$ influx effectively limits the lifetime of active TCR microclusters by increasing the rate at which they are transported to the synapse center, where signaling is terminated (*Yu et al., 2010*; *Varma et al., 2006*; *Yokosuka et al., 2005*). In addition, the accumulation of STIM1 and Orai1 in the ADZ reveals a new type of CRAC channel self-organization. At the level of single ER-PM junctions, STIM1 and Orai1 complexes self-organize through a diffusion trap mechanism based on STIM1 binding to the PM and Orai1 (*Wu et al., 2014*). At the synapse CRAC channels self-organize in a second way, by promoting the retrograde flow of actin that concentrates ER-PM junctions and CRAC channels in the ADZ. Given evidence that $Ca^{2+}$ locally regulates exocytosis in T cells and mast cells (*Pores-Fernando and Zweifach, 2009*; *Holowka et al., 2012*) CRAC channel self-organization may ensure that $Ca^{2+}$ is optimally positioned to serve critical $Ca^{2+}$-dependent functions including the directional secretion of cytokines like interleukin 2 and inter-feron-γ (*Huse et al., 2006*) that drive subsequent phases of the immune response.

## Materials and methods

### Cells and reagents

Cells were cultured at 37°C in a humidified incubator with 5% $CO_2$. Jurkat E6.1 cells (ATCC) were maintained in RPMI 1640 supplemented with 1% L-alanyl-glutamine and 10% fetal bovine serum (all from Gemini Bioproducts, West Sacramento, CA). Primary human peripheral blood CD4[+] T cells were obtained without donor identifiers from the University of Pennsylvania's Human Immunology Core under an Institutional Review Board approved protocol. Lymphoblasts were generated by

stimulating primary T cells for 24 hr with human T-Activator CD3/CD28 magnetic beads (Dynabeads, Life Technologies) and cultured in RPMI 1640 supplemented with 1% GlutaMAX, 1% penicillin-strep-tomycin (all from Invitrogen, Carlsbad, CA) and 10% fetal bovine serum (Atlanta Biologicals, Nor-cross, GA) prior to lentiviral transduction. Sulfinpyrazone, (-)-blebbistatin, and 2-APB were from Sigma-Aldrich (St. Louis, MO), and fura-2/AM was from Invitrogen. IL-2 was obtained through the AIDS Research and Reference Reagent Program, Division of AIDS, National Institute of Allergy and Infectious Diseases, National Institutes of Health; human rIL-2 was from M. Gately, Hoffmann-LaRoche, Nutley, NJ.

## Plasmids and transfection

Cloning of mCh-STIM1 was as described (*Luik et al., 2006*). Orai1-EGFP was a gift from T. Xu (*Xu et al., 2006*), F-tractin-P-tdTomato was a gift from J.A. Hammer III (*Yi et al., 2012*), PAGFP-actin was a gift from C.G. Galbraith (*Galbraith et al., 2007*), GFP-actin was from Clontech (Mountain View, CA), and ER-GFP (GFP-17) was a gift from N. Borgese (*Bulbarelli et al., 2002*). ER-mCh was made using site-directed mutagenesis to introduce a Not1 restriction site after GFP in GFP-17 (pri-mers: 5'GAT GAA CTA TAC AAA GCG GCC GCT GAG CAG AAG CTG ATC T 3' and reverse com-plement), then cloning mCherry into Kpn1/Not1 sites of the resulting plasmid. EB1-EGFP was a gift from L. Cassimeris (Addgene plasmid #17234; *Piehl and Cassimeris, 2003*) and EGFP-MyH9 was a gift from R.S. Adelstein (Addgene plasmid #11347; *Wei and Adelstein, 2000*). EGFP-Abi1 was cloned by digesting Abi1 from p-EYFP-Abi1 (gift from A.M. Pendergast; *Courtney et al., 2000*) and cloning into *Bgl*II site of p-EGFP-C1 (Clontech, Mountain View, CA). cDNA encoding Lifeact-GFP was a gift from R. Wedlich-Soldner (*Riedl et al., 2008*) and was subcloned into pDONR221 and sub-sequently into the lentiviral expression vector pLX301 using Gateway Technology. Jurkat cells at a density of 4–6 x $10^7$/ml in Ingenio electroporation solution (Mirus Bio LLC, Madison, WI) were elec-troporated in 0.4 cm cuvettes with 6–20 μg of plasmid DNA 40–48 hr prior to imaging.

## Primary T lymphoblast transduction

Primary human T lymphoblasts were transduced with Lifeact-GFP lentivirus 24 hr after stimulation. Lentivirus and 8 μg/ml polybrene (Sigma-Aldrich) were mixed with 5–10×$10^6$ T cells in 5 ml round bottom polystyrene tubes and centrifuged at 1,200 g for 2 hr at 37°C. Lentivirus-containing medium was then replaced with primary human T cell culture medium, and the cells were returned to the incubator. Two days after transduction, the medium was supplemented with 2 μg/ml puromycin, and cells were cultured for an additional four days before magnetic removal of the activator beads. Cells were cultured for an additional 1–2 days in medium with 2 μg/ml puromycin and 10 U/ml IL-2 before use (day 8–9 after activation).

## Cell stimulation and solutions

Stimulatory coverslips were washed with 100% ethanol, then coated overnight at 4°C with 10 μg/ml monoclonal anti-CD3 (OKT3 from eBiosciences, San Diego, CA for Jurkat cells and from BioXCell, Lebanon, NH for primary T lymphoblasts) in PBS and washed thoroughly with PBS. Where indicated, coverslips were subsequently coated with 2 μg/ml human ICAM-1 Fc chimera (R&D Systems, Minne-apolis, MN) for 2 hr at 20–22°C then washed thoroughly with PBS. Unless otherwise noted, cells were stimulated on the microscope at 37°C in Ringer's solution containing (in mM): 155 NaCl, 4.5 KCl, 2 CaCl$_2$, 1 MgCl$_2$, 10 D-glucose and 5 Na-HEPES (pH 7.4). In solutions with >2 mM CaCl$_2$, [NaCl] was reduced to maintain normal osmolarity, and in solutions with <2 mM Ca$^{2+}$, MgCl$_2$ was substituted for CaCl$_2$. In Ca$^{2+}$-free Ringer's solution, 1 mM EGTA and 2 mM MgCl$_2$ were substituted for CaCl$_2$. In Ca$^{2+}$ imaging experiments, all solutions contained 250 μM sulfinpyrazone to inhibit fura-2 extrusion. Cell imaging commenced within 3–7 min after loading cells onto coverslips, and all images were collected from cells having a constant, maximal diameter, a ruffling edge and in cells expressing fluorescently labeled actin or F-tractin-P, a clearly defined actin ring with retrograde flow in Ca$^{2+}$-containing Ringer's solution. [Ca$^{2+}$]$_i$ sometimes failed to decline to a minimum baseline level following Ca$^{2+}$$_o$ removal (described in *Figure 5*), and the persistent [Ca$^{2+}$]$_i$ elevation was associated with actin treadmilling and a ruffling lamellipod. Therefore, in the experiments of *Figures 4A*, *6* and *7* we limited our analysis to cells that lost the ruffling lamellipod and retrograde flow upon Ca$^{2+}$$_o$ removal.

## TIRF imaging and photoactivation

In *Figure 1* and *Figure 2—figure supplement 1*, TIRF images were acquired at 32–37°C on a custom-built through-the-objective TIRF microscope using an Axiovert S100TV base and a Fluar 100X, 1.45 NA oil-immersion objective (Carl Zeiss, Oberkochen, Germany). For simultaneous acquisition of GFP and mCherry/tdTomato, a Di01-R488/561 dual-band dichroic mirror (Semrock, Rochester, NY) directed excitation light from Sapphire 488-nm and Compass 561-nm lasers (Coherent, Santa Clara, CA) to the cells. Two bands of fluorescence emission were collected onto an Andor iXon DU897E EMCCD camera using an Optosplit-II (Cairn Research, Kent, UK) image splitter containing a dichroic mirror (FF580-FDi01, Semrock) and emission filters for GFP (FF02-525/50, Semrock) and mCherry/tdTomato (E600LP, Chroma, Bellows Falls, VT). Laser shutters and image acquisition were controlled by Micro-Manager (*Edelstein et al., 2010*).

All other TIRF and photoactivation experiments (*Figure 2–7* and associated supplements) were performed in the Stanford Cell Sciences Imaging Facility on a Nikon Eclipse-TI inverted microscope platform with a PLAN APO-TIRF 100X 1.49 N.A. oil-immersion objective, an environmental chamber for acquisition at 37°C, and a Perfect Focus System (Nikon, Tokyo, Japan). Images were collected with a Neo sCMOS camera (Andor, Belfast, UK), with 2x2 binning for photoactivation experiments only. A Lambda XL lamp and Lambda 10–3 filter wheel (Sutter, Novato, CA) were used for widefield epifluorescence illumination, and 488- and 561-nm lasers were used for through-the-objective TIRF imaging of GFP and tdTomato, respectively. GFP was imaged using a TRF49904-ET-488-nm laser bandpass filter set, while F-tractin-P-tdTom was imaged using a TRF49909-ET-561-nm laser bandpass filter set (Chroma). For all photoactivation experiments, a constant exposure time and illumination intensity was used, and photobleaching was less than 10% over the duration of each experiment. For photoactivation of PAGFP, a Mosaic digital illumination system (Andor) was used to steer a 405-nm laser to a user-defined region on the coverslip and photoactivate for 100 ms. All equipment was controlled using NIS-Elements software (Nikon).

## TIRF and calcium imaging

Transfected cells were loaded with 2.5 μM fura-2/AM at 22–25°C for 30 min in RPMI 1640 without phenol red or sodium bicarbonate. After washing, cells remained for 30 min in RPMI before they were resuspended in Ringer's solution immediately prior to loading onto coverslips. Cells were stimulated on anti-CD3 in 0.5 mM $Ca^{2+}_o$, followed by either $Ca^{2+}_o$ removal or application of 100 μM 2-APB in 0.5 mM $Ca^{2+}_o$, then perfusion with 2, 5, or 10 mM $Ca^{2+}$. F-tractin-P-tdTom and fura-2 were imaged on a Zeiss Observer Z1 inverted microscope at 37°C using an αPlan-Apochromat 100X, 1.46 N.A. oil immersion DIC objective (Carl Zeiss). Fura-2 imaging was performed using a Lambda XL lamp (Sutter), 380/15 and 357/10 excitation filters, 400-nm dichroic and 480-nm long pass emission filter (Omega Optical, Brattleboro, VT). Data are displayed as the ratio of emissions in response to excitation at 357 and 380 nm (357/380 ratio). F-tractin-P-tdTom images were acquired using through-the-objective TIRF with 561-nm laser excitation and a Zeiss 74HE filter set and an ImagEM-1K EMCCD camera (Hamamatsu, Hamamatsu City, Japan), and all equipment was controlled using Zeiss Axiovision software.

## Spinning disk confocal microscopy

Confocal imaging of Jurkat cells was performed in the Stanford Cell Sciences Imaging Facility on a Nikon Eclipse-TI inverted microscope platform with a CFI Plan Apochromat λ 60X, 1.4 N.A. oil-immersion objective, a CSU-X1 spinning disk (Yokogawa, Tokyo, Japan), an environmental chamber for acquisition at 37°C, and a Perfect Focus System (Nikon). Cells were illuminated with a 561-nm laser (Spectral Applied Research, Ontario, Canada) and images were projected via a 405/488/568/647 dichroic mirror and a 600/37 emission filter (Semrock) to an iXon Ultra 897 EMCCD camera (Andor). All equipment was controlled using NIS-Elements software.

Primary human CD4[+] lymphocytes expressing Lifeact-GFP were imaged on an Axiovert 200M microscope (Carl Zeiss) equipped with a spinning disk confocal system (Yokogawa), a 63x Plan Apo, 1.4 N.A. oil immersion objective and an environmental chamber for acquisition at 37°C. Cells were illuminated with a 488 nm laser (Melles Griot, Carlsbad, CA) and images were projected via a 405/488/561/640 dichroic mirror and a 527/55 emission filter (Chroma) to an Orca ER CCD camera (Hamamatsu). Sets of 3 image planes collected at 0.25 μm increments were collected every 2 s and

displayed as maximum intensity projections. All equipment was controlled using Volocity v. 6.3 imaging software (Perkin Elmer, Waltham, MA).

## Immunocytochemistry

Jurkat T cells were allowed to settle onto stimulatory coverslips in 0.5 mM $Ca^{2+}$ Ringer's solution at 37°C for 4 min, then transferred to either 0 or 0.5 mM $Ca^{2+}$ for 2.5 min before fixation with 4% paraformaldehyde in 10 mM PBS for 20 min. Cells were washed with 10 mM PBS containing 50 mM glycine, then permeabilized for 5 min with 0.1% Triton X-100 and blocked for 1 hr in 10 mM PBS, 50 mM glycine, and 10% fetal bovine serum. Cells were incubated at 20-22°C with anti-WAVE2 (H-110, Santa Cruz Biotechnology, Dallas, TX) diluted to 4 μg/ml in blocking buffer for 1 hr followed by 1-hr incubation with 2 μg/ml Alexa Fluor-488 goat anti-rabbit secondary antibody and 0.2 units/ml Alexa Fluor-594 phalloidin (Thermo Fisher, Grand Island, NY). Cells were washed extensively with blocking buffer between incubations and imaged immediately by TIRF.

## Data analysis

All images were background-corrected and analyzed using ImageJ (*Schneider et al., 2012*). In kymographs, each pixel represents the average intensity across the 5-pixel width of the scan line. Spatiotemporal image correlation spectroscopy (STICS) analysis was used to determine the direction and velocity of actin movement by applying an ImageJ plugin (STICS map jru v2, developed by Jay Unruh at Stowers Institute for Medical Research in Kansas City, MO) based on methods developed by Hebert et al. (*Hebert et al., 2005*). Sample sizes were determined based on previous in situ measurements of actin depolymerization rates (*Theriot and Mitchison, 1991*) and live-cell imaging studies of the immune synapse (*Babich et al., 2012*). To measure the $[Ca^{2+}]_i$ dependence of F--tractin-P fluorescence, F-tractin-P intensity was normalized to an average of the final five images collected in 0 $Ca^{2+}$, at which time the fura-2 ratio had reached a minimum value of $0.39 \pm 0.04$ (mean ± SEM, n = 26 cells). Data points represent the mean F-tractin-P fluorescence and fura-2 ratio when both signals were at steady-state and the sample size was selected to ensure that the cell population represented a broad range of $[Ca^{2+}]_i$. PAGFP fluorescence decay was measured within an ROI encompassing the photoactivated region that was moved centripetally to remain centered on the fluorescent bar of actin. The time course was fitted by a single exponential function using IgorPro (Wavemetrics, Portland, OR). At each time point, the mean intensity across the width of the photoactivated bar was measured along a line perpendicular to the direction of movement, and displacement of the peak fluorescence value versus time was used to calculate velocity. To create fluorescence profile plots of PAGFP-actin incorporation, the fluorescence intensity as a function of radial position was calculated by averaging across 19 pixel-wide scan lines, subtracting the average of 10 scans acquired before photoactivation, and plotting the result relative to this pre-photoactivation signal.

## Acknowledgements

Special thanks to Nathan H Roy for the preparation of primary human T cells expressing Lifeact-GFP. We thank T Xu, JA Hammer III, CG Galbraith, N Borgese, L Cassimeris, RS Adelstein, R Wedlich-Soldner, and A Pendergast for generous gifts of plasmids used in this study. We thank We thank Julie Theriot for thoughtful discussions and for generously sharing her imaging system, and members of the Lewis lab for their helpful feedback. The Stanford Cell Sciences Imaging Facility Nikon Eclipse-TI microscope was funded by the Beckman Center and the Stanford School of Engineering. This work was supported by NSF-GRFP to CH and NIH grants R37GM45374 to RSL and R01GM104867 to JKB. The authors have no competing interests.

## Additional information

### Funding

| Funder | Grant reference number | Author |
| --- | --- | --- |
| National Science Foundation | Graduate Student Research Fellowship | Catherine A Hartzell |

| National Institutes of Health | R37GM45374 | Richard S Lewis |
| National Institutes of Health | R01GM104867 | Janis K Burkhardt |

The funders had no role in study design, data collection and interpretation, or the decision to submit the work for publication.

### Author contributions

CAH, Conception and design, Acquisition of data, Analysis and interpretation of data, Drafting or revising the article; KIJ, Acquisition of data, Analysis and interpretation of data; JKB, Conception and design, Drafting or revising the article; RSL, Conception and design, Analysis and interpretation of data, Drafting or revising the article

### Author ORCIDs

Richard S Lewis, http://orcid.org/0000-0002-6010-7403

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
