## [Decision Letter]

Thank you for submitting your article "Calcium influx through CRAC channels controls actin organization and dynamics at the immune synapse" for consideration by *eLife*. Your article has been favorably evaluated by Richard Aldrich (Senior editor) and three reviewers, one of whom, David Clapham, is a member of our Board of Reviewing Editors. The following individual involved in review of your submission has agreed to reveal their identity: Michael Cahalan (peer reviewer).

The reviewers have discussed the reviews with one another and the Reviewing Editor has drafted this decision to help you prepare a revised submission.

Summary:

Hartzell and Lewis examine the dynamics of Stim/Orai redistribution at the immunological synapse. The more specific question is how calcium entry through this channel alters actin and ER to drive specialization of the region (e.g., concentration of Orai Ca^2+^ channels, etc.) while recycling T cell receptors. The main conclusions are that actin reorganizes into a lamellipod with retrograde actin flow. Ca^2+^ influx promotes actin depolymerization and localizes actin polymerization to the periphery of the lamellipod. Overall, the continuous actin reorganization drives microtubule bound ER and to the synapse center, presumably keeping the ER/PM junctions formed by Stim intact.

All three reviewers agreed that the overall the work is of high quality, the movies fascinating and the conclusions well-supported. This insightful work makes an important contribution toward understanding the molecular dynamics of the immunological synapse, in relation to calcium signaling and actin cytoskeleton.

Essential revisions:

The major criticisms that require new experiments were:

1) It has been shown that the actin dynamics in Jurkat and primary mouse/human T cells differ dramatically (Kumari et al. *eLife* 2015) making it difficult to directly extrapolate the results from the Jurkat T cell system to the ontology of the immune synapse as a whole. As suggested above, validation of some key findings in a primary cell system would in my opinion add significance to this study. In doing so, the authors would also include conditions in which ICAM1 is present. WAVE2 has been shown to directly regulate ICAM1 and localize to ICAM1 high pSMAC. The inclusion of ICAM1 is important for a functional relevance of the mechanism, in vivo T cells will encounter antigen in the context of ICAM1 and if WAVE2 is recruited to LFA1-1 this will affect the interpretation of the specific recruitment of WAVE2 to the LM actin ruffles. In addition, the Kumari paper shows WAVE2 immuno staining that is relatively uniform across the synapse, which differs from the results presented here.

This would require experiments comparing what is observed in Jurkats to T cells. Specifically, the reviewer said "the experiments looking at synapses between primary T-APC can be done in 2 months."

---

## [Author Response]

*Essential revisions:*

*The major criticisms that require new experiments were:*

*1) It has been shown that the actin dynamics in Jurkat and primary mouse/human T cells differ dramatically (Kumari et al. eLife 2015) making it difficult to directly extrapolate the results from the Jurkat T cell system to the ontology of the immune synapse as a whole. As suggested above, validation of some key findings in a primary cell system would in my opinion add significance to this study. In doing so, the authors would also include conditions in which ICAM1 is present. WAVE2 has been shown to directly regulate ICAM1 and localize to ICAM1 high pSMAC. The inclusion of ICAM1 is important for a functional relevance of the mechanism, in vivo T cells will encounter antigen in the context of ICAM1 and if WAVE2 is recruited to LFA1-1 this will affect the interpretation of the specific recruitment of WAVE2 to the LM actin ruffles. In addition, the Kumari paper shows WAVE2 immuno staining that is relatively uniform across the synapse, which differs from the results presented here.*

*This would require experiments comparing what is observed in Jurkats to T cells. Specifically, the reviewer said "the experiments looking at synapses between primary T-APC can be done in 2 months."*

We agree that it is important to show that the Ca^2+^ effects we report here also extend to a more physiological synapse model. We have responded to the reviewers’ comments by conducting a new series of experiments using human CD4^+^ T lymphoblasts plated on anti-CD3 alone as well as anti-CD3 + ICAM1, in collaboration with Dr. Janis Burkhardt’s group. We find that Ca^2+^ removal slows actin flow and reduces the extent of the lamellipod in a reversible fashion. These effects occur whether or not ICAM1 is present. Thus, Ca^2+^ affects actin organization and dynamics in primary human T cells as well as Jurkat T cells, although the effects are somewhat less dramatic in primary cells. These new results demonstrate that the effects of Ca^2+^ are not specific to the Jurkat cell, and do extend to a more physiological synapse model. The new data are presented in Figure 3 and Video 7 and Video 8 and discussed in the last paragraph of the subsection “Calcium influx organizes actin at the synapse” and in the second paragraph of the Discussion.

We believe the different distribution of WAVE2 in our study and that of Kumari et al. is due to the fact that the localization of WAVE2 varies at different times during synapse development. In the Kumari paper WAVE2 localization (which was punctate throughout the cell footprint) was shown only in a single cell imaged at a very early time point (2’ after stimulation) when the cell was still in the spreading phase of synapse formation and the lamellipod had not yet formed (Figure 1—figure supplement 4B). In another study by Le Floc’h et al. (Le Floc’h et al. J. Exp. Med. 210:2721–37, 2013.), WAVE2 was also localized throughout the cell footprint at an early time, but became concentrated at the cell edge by 9 min, when the synapse had fully formed (see Figure 1 of that paper). At later times (18-45 min), WAVE2 became most concentrated in the lamella/pSMAC where LFA-1 accumulates. Our results are consistent with these studies, in that we are specifically visualizing WAVE2 at 6 min after plating, which is shortly after the cell has fully spread, and we verify this by selecting cells that have a clearly lamellipod by phalloidin staining. This phase of synapse formation is equivalent to the 9 min time point in Le Floc’h et al.